# Place-based approaches to improve health and development outcomes in young children: A scoping review

Fiona C. Burgemeister[1‡]*, Sharinne B. Crawford[1☯], Naomi J. Hackworth[1,2,3☯], Stacey Hokke[1☯], Jan M. Nicholson[1,3,4‡]

**1** Judith Lumley Centre, La Trobe University, Bundoora, Victoria, Australia, **2** Parenting Research Centre, East Melbourne, Victoria, Australia, **3** Murdoch Children's Research Institute, Parkville, Victoria, Australia, **4** Queensland University of Technology, Brisbane, Queensland, Australia

☯ These authors contributed equally to this work.
‡ FCB is the lead author. JMN is the senior author.
* f.burgemeister@latrobe.edu.au

**Data Availability Statement:** All relevant data are within the manuscript and its Supporting Information files.

## Abstract

This scoping review examines the strength of evidence for the effectiveness of public policy-led place-based initiatives designed to improve outcomes for disadvantaged children, their families and the communities in which they live. Study designs and methods for evaluating such place-based initiatives were assessed, along with the contexts in which initiatives were implemented and evaluated. Thirty-two reports relating to 12 initiatives were included. Eleven initiatives used a quasi-experimental evaluation to assess impact, although there were considerable design variations within this. The remaining initiative used a pre- and post- evaluation design. Place-based initiatives by definition aim to improve multiple and interrelated outcomes. We examined initiatives to determine what outcomes were measured and coded them within the five domains of pregnancy and birth, child, parent, family and community. Across the 83 outcomes reported in the 11 studies with a comparison group, 30 (36.4%) demonstrated a positive outcome, and all but one initiative demonstrated a positive outcome in at least one outcome measure. Of the six studies that examined outcomes more than once post baseline, 10 from 38 outcomes (26.3%) demonstrated positive sustained results. Many initiatives were affected by external factors such as policy and funding changes, with unknown impact on their effectiveness. Despite the growth of place-based initiatives to improve outcomes for disadvantaged children, the evidence for their effectiveness remains inconclusive.

## Introduction

Socio-economic disadvantage clusters within families and the areas where they live [1]. Disadvantage is becoming increasingly geographically concentrated [2, 3], with neighbourhood disadvantage exacerbating the challenges families face [2, 4] and contributing to intergenerational poverty. Place-based approaches for children include a locational element in

**Funding:** FB holds an Australian Government RTP PhD scholarship. The funders had no role in study design, data collection and analysis, decision to publish, or preparation of the manuscript.

addressing complex social and economic issues that impact adversely on the health and wellbeing of children and their families [3]. Such initiatives address not just child outcomes (e.g. academic, social-emotional, physical, cognitive), but also the parent (e.g., physical/mental health, education, employment), family (e.g., home learning environment, parenting style) and community (e.g., cohesion, safety, services) circumstances that impact on child trajectories [5]. The purpose of this review is to determine the strength of evidence for the effectiveness of initiatives that use a place-based approach to improve outcomes for children in their early years.

Place-based approaches target defined geographic areas and take an ecological perspective, addressing individual, family, organisational and community level issues. The approach tends to be participatory and tailored to local needs, delivered across multiple sites and involving multiple delivery organisations, with shared goals and funding [6]. Described as a 'multidimensional saturation model', place-based approaches are theorised to be advantageous as they "enable the targeting of people experiencing multiple and inter-related forms of disadvantage and provide a platform for the delivery of a more integrated and holistic suite of services and supports" [7 p21].

In the early 1990s, 'place-based' (also known as 'area-based' or 'neighbourhood-level') initiatives emerged in the United Kingdom (UK), Canada and the United States of America (USA) with the goal of improving multiple outcomes for children and their families [5]. Large, nation-wide flagship programs such as Sure Start Local Programmes (which evolved to become Children's Centres) [8] in the UK are well known and have been subject to intense scrutiny, while in the USA, successful local programs such as the Harlem Children's Zone have resulted in the development of nationally funded initiatives [9]. In Australia, the federal government introduced Communities for Children, which was modelled on Sure Start [10].

While many place-based initiatives globally have been established through community-led coalitions with philanthropic funding, governments have increasingly recognised their value, making them a core tenet of social and health equity policy [11, 12]. Such policy-led initiatives must find a balance between 'top-down', and 'bottom-up' approaches, whereby broad objectives are determined centrally ('top-down'), but addressed locally ('bottom-up') [6, 13]. A review of federal government place-based initiatives conducted by Wilks and colleagues [6] identified several elements common to many initiatives. Fig 1 presents a summary of these elements in relation to design, delivery and evaluation approaches.

The complex designs of place-based initiatives pose unique challenges for evaluation. It is difficult to develop and execute integrated measurement of broad top-down objectives, location-specific bottom-up objectives, as well as process, impact and economic measures. Much has been written about these evaluation challenges, either prospectively [13–15] or retrospectively [6, 16, 17]. Local evaluation, whereby each geographic area conducts its own discreet evaluation, is often part of the framework in large place-based initiatives, however integrating local evaluation 'learnings' that can be applied across the whole initiative has proven difficult [17]. This complexity is compounded by changing social, economic and political contexts that influence how initiatives are implemented and evaluated [18, 19].

There is no contemporary literature review that examines evidence of the effectiveness of place-based initiatives for children in their early years. Existing syntheses have included a narrative review [5], critical commentaries [20, 21], reviews that considered national level initiatives only [6, 21] or a single element of activity such as community involvement [22]. One review of place-based initiatives [23] had a broad, non-child specific focus and found weak evidence of effectiveness. We address the limited previous research in relation to child focused place-based initiatives by undertaking a scoping review. A scoping review approach enables a broad focus that encapsulates initiative design, study designs and methods used for evaluating child focused place-based initiatives, in addition to an examination of effectiveness [24].

* Local evaluation may be commissioned locally or be part of the independent, centrally managed whole-of-initiative evaluation

**Fig 1. Common elements in the design, delivery and evaluation of place-based initiatives for children.**

This review focuses on public policy-led place-based initiatives. In determining what meets the criteria for a 'place-based initiative', we have erred on the side of inclusion. Many place-based initiatives are labelled as such, and remain so for the life of the initiative. For others, the notion that risk and protective factors are spatially differentiated and that disparities in outcomes varies between neighbourhoods informs their design and delivery, irrespective of the number of geographic areas targeted or the mechanisms by which the geographic areas were chosen. Some initiatives commence in a defined set of localities, then rapidly expand to cover numerous localities due to their perceived success, and some USA initiatives involved every county within a state. They remain place-based in their approach to design and delivery (e.g., local needs require local solutions), and their underlying aim is to reduce the inequality gap between the children and families in their population of interest compared to the rest of the country. For the purpose of this review, we have included these initiatives.

This review focuses on early childhood initiatives that target (but are not necessarily limited to) pregnancy to four years. Children's health and development outcomes are influenced by their experiences early in life [25–27]. The impact of socioeconomic disadvantage starts before a child is born, and inequalities are apparent from the earliest years [28, 29]. Interventions in the first three years of a child's life, combined with high quality childcare and preschool (kindergarten) have been shown to be effective at reducing the inequality gap [30].

The aims of the review are to identify:

1. Study designs and methods used in evaluating public policy-led place-based initiatives aiming to improve outcomes for young children, their families and the communities in which they live;

2. The nature of the contexts in which these place-based initiatives have been implemented and evaluated; and

3. The strength of evidence for the effectiveness of place-based initiatives.

## Methods

A scoping review was informed by Peters and colleagues' guidance on conducting systematic scoping reviews [24] and reported in accordance with the PRISMA-ScR guidelines [31] (see S1 Checklist).

### Information sources

**Database search.**  Two database searches were conducted, one in August 2016 with no date restrictions, and repeated in July 2020 for the time period September 2016 to July 2020 with the following search criteria. English-language articles were searched in CINAHL, Pro-Quest Central, SCOPUS, Informit (all databases) and Embase. Five categories of search terms were combined (sample search strategy provided in S1 Appendix): 1. Child, parent, family; 2. Place-based/level, area-based/level, community-based/level, neighborhood-based/level, complex community, collective impact; 3. Disadvantage, poverty, vulnerable, socio-economic, inequality, well-being; 4. Intervention, initiative, program, trial; and 5. Outcome, impact, efficacy, evaluate, feasibility, protocol, pilot. Additional papers were retrieved by examining reference lists of identified papers and by separate searches using the titles of identified place-based initiatives.

**Grey literature search.**  Many evaluations of public policy driven place-based initiatives are commissioned to consultants, independent research groups, research consortiums or university departments and are presented in report form. Inclusion of material not controlled by commercial publishers ("grey literature") in evidence reviews reduces publication bias and provides a more complete and balanced picture of the evidence [32]. We used three approaches to identify grey literature relevant to this review: 1. A Google search of known initiatives and initiatives identified via secondary sources, with the terms 'evaluation', 'report' or 'pdf' entered in an attempt to source evaluation reports; 2. Searching known databases containing research and evaluation reports (e.g., www.childtrends.org, www.researchconnections.com and Child Family Community Australia Information Exchange); and 3. Searching websites established specifically for initiatives and/or the initiative's evaluation (e.g., National Evaluation of Sure Start website and Toronto First Duty website).

### Eligibility criteria

**Types of studies.**  We included initiatives if an impact evaluation study had been conducted. All types of impact study designs were considered eligible for inclusion (e.g., randomised controlled trials (RCTs), quasi-experimental, non-experimental, cohort, cross-sectional, pre- and post-), if at least one child outcome had been reported.

**Types of place-based initiatives.**  *Inclusion criteria*. Literature pertaining to a place-based initiative was initially included if the initiative met the following criteria:

• Population: targeted (but not limited to) children (infancy to 4 years) and pregnant women who live in socioeconomically disadvantaged areas.

- Place-based. Showed evidence of a place-based approach, with a focus on people and place [33].

- Location: high income countries (as defined in NationMaster) [34].

- Sponsoring organisation: government administered program. Showed evidence of federal or state government initiating, leading and/or managing the initiative.

- Size/scale of initiative: implemented at a national, state or regional level, or was a multi-site demonstration project.

- Outcomes: goal of improving multiple outcomes for children and their families.

*Exclusion criteria*. Initiatives were excluded if the primary goal was improving a single child outcome domain (e.g., obesity prevention, prevention of child abuse/neglect), targeted a specific adult or child clinical population, or if the primary aim was broad social, health, economic, or physical regeneration or improvement (e.g., the physical quality of homes or public spaces), even though a subsidiary benefit may have been improved outcomes for children.

**Selection of sources of evidence.** *Inclusion criteria*. Article title and abstract screening was initially conducted by Author 1 (FB) with potentially eligible studies included for full text review. Author 1 and Author 5 (JN) conducted the full text review, with disagreements resolved through consensus. In this review, multiple results from the same initiative are reported together. Therefore, once initiatives were selected for inclusion, publications that presented results from the same initiative were collated and assessed as 'primary' or 'secondary' studies. Primary studies were those that provided the principal source of outcomes evaluation information for each initiative for completion of the evidence appraisal. Secondary studies were those that provided detail about process evaluation and contextual information about how the intervention changed over time, and were included in the review only where this information was not available in the primary source. Many of the initiatives reported impact evaluations conducted at multiple time points. In these cases, the most recent was used as the primary source, and supplemented with the earlier reports as required. For some initiatives, evaluations were reported in both peer reviewed and grey literature. Peer reviewed papers were prioritised for inclusion over grey literature where they were reporting on the same data.

*Exclusion criteria*. Articles were excluded if they reported no original data or evaluated only a single component of a broader place-based initiative, including local evaluations.

**Types of outcome measures and other data items of interest.** Place-based initiatives by definition aim to improve multiple and interrelated outcomes across pregnancy and birth, child, parent, family and community domains [35]. Rather than approaching this scoping review with a pre-determined set of outcomes, we examined the included initiatives to determine what outcomes were measured and collated and coded them as per the domains and categories in Table 1. In determining whether the place-based initiatives were effective at improving outcomes (Aim 3), significance was set at $P \leq 0.05$.

Other data items of interest were broadly informed by our research questions and are summarised in Table 1. Where appropriate, the beginning of each sub-section briefly defines and justifies the inclusion of the item of interest. We collected overview data to enable the characteristics of the initiatives to be described (location, size/scope, year of commencement), along with initiatives' aim and service model, funding and delivery structure, the size and selection process for local delivery areas, and theories of change. These were summarised and combined with outcome data to help shed light on what aspects may contribute to effectiveness. As our

**Table 1. Data items applied to initiatives.**

| Item | Data items and categories |
|---|---|
| Initiative name | Free text |
| **Characteristics of initiative** | |
| Description | Free text. Brief descriptive overview of initiative, including aim, service model, funding, delivery structure. |
| Location | Free text. Includes: name of country, whether national or state initiative, name of state (where applicable), number of locations initiative was implemented (where available) |
| Size of delivery area | Free text |
| Spatial targeting | Free text |
| Theory of change | Free text summary; mechanisms by which initiative would improve outcomes |
| Time-limited or ongoing | Time-limited; ongoing |
| Stage of intervention at time of last evaluation | No. of years |
| Evaluation before or at time of implementation | Yes; no |
| Peer reviewed or grey literature | Peer reviewed only; grey only; mixed |
| Who was intervention targeting | Free text. Includes: age range of children, whether families and communities were targeted |
| **Context in which initiative was implemented and implemented** | |
| Context | Free text. Descriptive overview of the context in which the initiative was delivered and evaluated |
| How environment affected initiative | 1 Initiative funding changes 2 Initiative scope changes 3 Initiative design changes 4 Broader policy impacts on population behaviour 5 Evaluation funding/scope changes 6 Unknown/unclear 7 None |
| **Evaluation design** | |
| Evaluation design (in addition to impact study) | 1 Process evaluation 2 Local evaluation 3 Economic/cost-effectiveness evaluation |
| What did process evaluation measure and how | Free text |
| Impact study design | 1 RCT 2 Quasi-experimental 3 Cross-sectional 4 Cohort 5 Pre- & Post- 6 Longitudinal 7a Population sample–general 7b Population sample–intervention areas 8 Intervention sample 9 Time series |
| Level of evidence (NHMRC) | I Systematic review of all relevant RCTs |
| | II Properly designed RCT |
| | III-1 Well designed pseudo-RCT |
| | III-2 Comparative studies (or systematic reviews of such studies) with concurrent controls and allocation not randomised, cohort studies, case-control studies, or interrupted time series with a control group |
| | III-3 Comparative studies with a historical control, two or more single arm studies, or interrupted time series without a parallel control group |
| | IV Case series, post-test or pre-test/post-test with no control group |
| Clear description of methodology | Yes; partly; no |
| Data collection methods | 1 Face-to-face interviews 2 Telephone interviews 3 Child/family assessments 4 Self-administered survey 5 Routinely collected datasets |
| Study sample | Free text |
| Length of study/Years of study | No. of years, Years |
| Quality rating based on fit-for-purpose | High; medium; low |
| **Outcomes** | |

(*Continued*)

**Table 1.** (Continued)

| Item | Data items and categories |
|---|---|
| Outcome domains | **A Pregnancy & birth** |
| | A1 Birthweight & age |
| | A2 Pregnancy/delivery |
| | A3 Prenatal & infant health |
| | A4 Type of feeding and duration of breastfeeding |
| | **B Child/youth** |
| | B1 Physical health |
| | B2 Emotional and behavioural functioning |
| | B3 Temperament/self-regulation |
| | B4 Attendance at formal childcare/early learning |
| | B5 Developmental status |
| | B6 School readiness |
| | B7 Educational attainment & attendance |
| | B8 Language/cognition |
| | **C Parent** |
| | C1 Physical health status |
| | C2 Mental health status |
| | C3 Health risk behaviours |
| | C4 Social support (personal) |
| | C5 Employment status/movement off benefits |
| | **D Family** |
| | D1 Parenting style |
| | D2 Partner relationship |
| | D3 Reading with child |
| | D4 Activities with child |
| | D5 Other family functioning |
| | D6 Household safety |
| | **E School/community** |
| | E1 Community involvement (eg volunteering, coaching) |
| | E2 Social cohesion/belonging |
| | E3 Neighbourhood safety |
| | E4 Service use (incl health, development, family support, childcare, early learning & schools) |
| | E5 Service quality (incl health, development, family support, childcare, early learning & schools) |
| | E6 Service availability/access |
| | E7 Child friendly community |
| Outcomes | +ve, -ve effect (P≤ .05), sustained (if multiple time points measured) Yes, or No/weak effect |

first aim was to examine the study designs and methods for evaluating place-based initiatives, we identified the following data items of interest: quality, overall evaluation design, length and timing, process evaluation, local evaluation, and impact study design. For impact study design we documented a range of design features including the study sample, comparison group (if relevant), and method of data collection. We were interested in the context in which initiatives were implemented and evaluated (Aim 2), therefore we initially summarised these findings in a free text field then specifically coded a range of items where the contextual environment directly affected the initiative (e.g., change in scope or funding).

## Data charting process

To extract key information on each initiative *A Schema for Evaluating Evidence on Public Health Interventions* [36] was used. This comprehensive framework for appraising evidence on public health interventions summarises evaluation design, the setting in which the intervention was implemented and evaluated, and outcomes. It has been used in a previous literature review of place-based interventions for reducing health inequalities [23].

To enable the Schema to be applied to each initiative, the following steps were taken. First, articles for each initiative were collated and identified as: 'primary outcomes paper', 'process evaluation paper'; or 'secondary study'. Using a template based on the Schema adapted to the current review aims, data were extracted from the collated articles and summarised in three databases: 1. Initiative description, context and implementation, 2. Study design and outcomes, and 3. Evaluation design. Data were coded where possible for ease of comparison. The data items and ratings categories used to populate these databases are provided in Table 1.

To assess data quality for each initiative, a quality assessment rating tool was developed. Drawing on evaluation methods typically used for place-based initiatives, combined with commentaries regarding the challenges and limitations of place-based initiative evaluations [13, 15, 23], we identified the following seven criteria as indicative of an appropriate fit for place-based initiative evaluations:

1. Included a broad range of outcome measures across child, family and community domains (assessed as Yes, Somewhat, No)

2. Measures were a good match for the stated outcomes for the initiative (Yes, Somewhat, No)

3. Evaluation was designed before or at the time of implementation (Yes, No, Unclear)

4. Evaluation allowed time for full implementation of the initiative (Yes, No)

5. Multiple impact time points were measured (Yes, No)

6. Change was measured at the population level (Yes, No)

7. Comparison group was appropriate (Yes, Partly, No, Not applicable)

Summarising of data and the quality ratings assessment were initially undertaken by Author 1 (FB), and databases were independently validated by Author 5 (JN). Where there was disagreement, consensus was reached through discussion. Meta-analysis of the data was not appropriate due to the heterogeneity of the outcomes, initiatives and population groups. Narrative summary was used to describe key findings for each research aim.

## Results

The original keyword database search conducted in August 2016 identified 2839 articles. Database searching using known place-based initiatives titles, hand searching reference lists and a search of the grey literature produced an additional 143 records. Following title and abstract screening, 1534 articles were excluded. The majority were excluded due to the search term 'community-based' identifying non-relevant articles (e.g., community-based HIV programs in Africa, community-based pediatric dental programs). Other common reasons for exclusion at this stage were: the initiative focussed on adults; was not place-based; and/or was not in a high income country. Full text screening for eligibility was undertaken on 92 records. This resulted in 31 reports that met all inclusion criteria, representing 11 initiatives.

The updated keyword database search conducted in July 2020 identified 2846 articles. An additional three articles were identified by hand searching reference lists. Following title and

abstract screening, 1781 articles were excluded. Full text screening for eligibility was undertaken on 57 records. This resulted in one additional article/initiative that met all inclusion criteria. When both the original and updated search findings were combined, 32 reports met all inclusion criteria, representing 12 initiatives. This process is represented in Fig 2 below.

## Characteristics of included studies

Of the 12 initiatives included for analyses, there were five national initiatives: one in Australia, Communities for Children [10, 37–39]; one in Ireland, the Area Based Childhood (ABC) Programme [40]; and three in the UK, Sure Start [8, 41–46], Neighbourhood Nurseries Initiative [47] and Flying Start [48–51]. There were four state or regional initiatives: one in Australia, Best Start [52–54]; and three in the USA, First Steps (First Steps) to School Readiness [55, 56], Smart Start [57, 58] and Georgia Family Connection [59]. The remainder were national or state demonstration projects which were smaller in scope: one in Canada, Toronto First Duty [60–63]; one in Ireland, National Early Years Access Initiative (NEYAI) [64]; and one in Scotland, Starting Well [65, 66]. Five initiatives commenced between 1990 and 2000 (Sure Start [8, 41–46], First Steps [55, 56], Smart Start [57, 58], Georgia Family Connection [59], Starting Well [65, 66]); five between 2001 and 2009 (Communities for Children [10, 37–39], Neighbourhood Nurseries Initiative [47], Flying Start [48–51], Best Start [52–54], Toronto First Duty [60–63]); and two after 2010 (ABC Programme [40], NEYAI [64]). Key characteristics of the 12 included initiatives are summarised in Table 2 and the initiatives are described in Table 3.

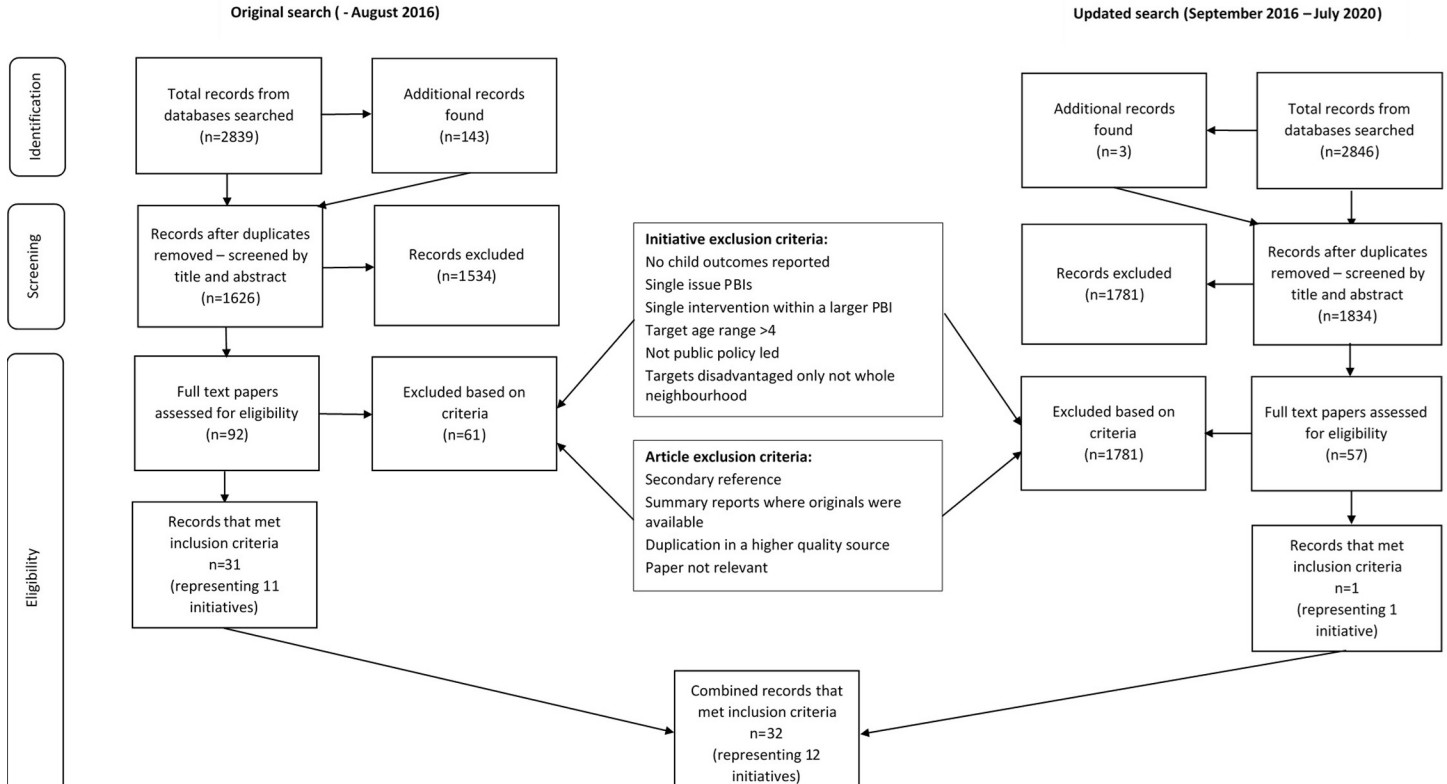

**Fig 2. Selection of articles for review of place-based initiatives to improve outcomes for children from disadvantaged backgrounds.**

**Table 2. Summary of included initiatives (n = 12).**

| Characteristics of included initiatives | | No. |
|---|---|---|
| Year initiative commenced | | |
| | 1990–2000 | 5 |
| | 2001–2009 | 5 |
| | 2010 - | 2 |
| Country | | |
| | Australia | 2 |
| | Canada | 1 |
| | Ireland | 1 |
| | United States of America | 3 |
| | United Kingdom | 5 |
| Time-limited or ongoing | | |
| | Time-limited | 4 |
| | Ongoing (in some form) | 8 |
| No. initiatives assessing[1] | | |
| | Pregnancy & birth outcomes | 3 |
| | Child outcomes | 12 |
| | Parent outcomes | 5 |
| | Family outcomes | 8 |
| | School and community outcomes | 5 |
| No. outcome domains assessed (range 1 to 19) | | |
| | 1 | 2 |
| | 2–5 | 6 |
| | 6–10 | 1 |
| | 11–19 | 3 |
| Evaluation framework (additional to impact study) [1] | | |
| | Process | 10 |
| | Local evaluation | 8 |
| | Cost-effectiveness/economic | 6 |
| Impact study design[1] | | |
| | Quasi-experimental | 11 |
| | Pre- & post- | 6 |
| | Longitudinal | 5 |
| | Time series | 1 |
| | Cross-sectional | 6 |
| | Cohort | 6 |
| | Population sample–general | 6 |
| | Population sample–intervention areas | 3 |
| | Intervention sample | 6 |
| | Purpose designed study sample | 8 |
| | Routinely collected data study sample | 6 |
| Context/changing environment impact[1] | | |
| | Intervention funding changes | 4 |
| | Intervention scope changes | 3 |
| | Intervention design changes | 6 |
| | Broader policy impacts on study population | 4 |
| | Evaluation funding/scope changes | 3 |
| Literature type, by initiative | | |

(*Continued*)

**Table 2.** (Continued)

| Characteristics of included initiatives | | No. |
|---|---|---|
| | Peer reviewed only | 1 |
| | Grey only | 5 |
| | Mixed | 6 |

[1] Initiatives may be counted more than once.

## Overview of initiatives

**Aims and service model.** A brief description of each initiative, including the aim and service model was extracted and is summarised in Table 3. There was considerable diversity in the aims of the initiatives and thus in the range of programs and services provided. Some focused primarily on strengthening universal services through 'joined-up working' and service integration (ABC Programme [40], Sure Start [8, 41–46], Best Start [52–54], Toronto First Duty [60–63], Flying Start [48–51], Starting Well [65, 66]), or on improving childcare and kindergarten quality (First Steps [55, 56], Smart Start [57, 58], NEYAI [64]). Others focussed more on addressing gaps in current service delivery (Communities for Children [10, 37–39], Georgia Family Connection [59], Neighbourhood Nurseries Initiative [47]). Models of service delivery also varied. Some initiatives provided centre-based delivery via children's centres (Neighbourhood Nurseries Initiative [47], Toronto First Duty [60–63]), others had a more diffuse model of service delivery in the community (ABC Programme [40], Communities for Children [10, 37–39], Georgia Family Connection [59], Starting Well [65, 66]), and some provided a mix of both.

**Funding and delivery structures.** Funding and delivery structures for all included initiatives were also extracted (not reported in tables for brevity). Some initiatives were wholly funded and implemented by government organisations (Sure Start [8, 41–46], Best Start [52–54], Flying Start [48–51], Starting Well [65, 66]). Others were funded by the government but contracted non-government organisations to deliver at the community level (Communities for Children [10, 37–39]). For Neighbourhood Nurseries Initiative [47], funding was available to both non-government and privately operated childcare centres. In Ireland, Canada and the USA it was more common for the government to work in partnership with philanthropic and corporate partners with shared responsibilities for funding, governance and implementation (ABC Programme [40], NEYAI [64], First Steps [55, 56], Smart Start [57, 58], Toronto First Duty [60–63], Georgia Family Connection [59]).

**Size and selection of delivery areas.** Previous research has highlighted the importance of geographic scale and the concept of 'place' as potential influences on the effectiveness of place-based initiatives [7, 23]. We extracted the size of local delivery areas and how they were selected, as summarised in Table 3. These varied considerably between initiatives and indeed was not uniform within initiatives. 'Place' in USA state-based initiatives (First Steps [55, 56], Smart Start [58, 67], Georgia Family Connection [59]) was defined at county level, and usually started as demonstration projects in a defined number of counties before expanding to cover the whole state. For the majority of the UK initiatives, areas were much smaller. Sure Start areas, for example, averaged around 13,000 people with around 700 children aged 0–3 and were targeted to 20% of the most deprived areas in England [8]. Flying Start targeted highly concentrated pockets of disadvantage within already deprived Local Authority areas, and used school catchment areas to define their delivery boundaries [48]. Toronto First Duty in Canada also based their delivery areas around schools, in keeping with their school hub service model

**Table 3. Overview and characteristics of included initiatives (n = 12).**

| First author(s), Year | Initiative | Description | Size of local delivery area | Spatial targeting | Theory of change | Target age of children | Ongoing or time-limited | Time between implementation & last outcomes evaluation | Overall evaluation designs[1] | Impact study design[2] | Level of evidence | Contextual factors influencing intervention/ evaluation[3] | No. of domains measured | No. of outcomes measured | Quality rating |
|---|---|---|---|---|---|---|---|---|---|---|---|---|---|---|---|
| **National Initiatives** | | | | | | | | | | | | | | | |
| Katz, 2007 [37] Edwards, 2009 [38] Edwards, 2011 [10] Edwards 2014 [39] | **Communities for Children (Australia)** | Initially implemented in 45 disadvantaged locations across Australia (now 52) from 2006. Services worked together to deliver a range of programs and services determined by local communities to address unmet needs. | Each CfC site was defined differently. Some sites were defined as one or more suburbs or postcodes and others were defined as one or more Statistical Local Areas or Collection Districts. | Sites were chosen in metropolitan, regional and remote locations that met the criteria for multiple aspects of disadvantage. | NGOs will **work with the community to identify and provide services to meet community need.** Service effectiveness is dependent not only on the nature and number of services, but also on **coordinated service delivery.** Child-friendly communities will be created that will lead to better outcomes for children and parents. | 0–12 (initially 0–5) | Ongoing | 5 years | 1–3 | 2, 4, 6, 7a | III-2 | 2, 3, 4, 5 | 4 | 14 | High |
| Hickey, 2018 [40] | **Area Based Programme (Ireland)** | Implemented in 13 disadvantaged communities across Ireland from 2013. It aimed to improve outcomes for children and parents, as well as support increased interagency working and embedding of evidence-based/ informed interventions in mainstream services. | Areas that were serviced by a consortium of service partners within a bounded geographical area (appears to be mostly suburb/ town) | Sites were chosen by application, according to guidelines which defined an area with as a geographic territory in which the resident population identified with each other as a community. Other criteria included: evidence of local need, quality of proposal, ability to capture local outcomes, and sustainability post funding | Break the cycle of child poverty through **integrated and effective services/ interventions** in the areas of child development, child well-being, parenting and educational disadvantage. **Building upon existing services,** improving planning and delivery of services and **embedding evidence-based programs in mainstream services through** greater integration, more **effective interagency working and involvement of local communities** will improve outcomes for children, families and communities. | Pregnancy– (age not specified) This evaluation examined changes for children ≥3 | Ongoing | 3 years | 1, 3 | 5, 8 | IV | 7 | 2 | 5 | Low |

*(Continued)*

**Table 3.** (Continued)

| First author(s), Year | Initiative | Description | Size of local delivery area | Spatial targeting | Theory of change | Target age of children | Ongoing or time-limited | Time between implementation & last outcomes evaluation | Overall evaluation designs[1] | Impact study design[2] | Level of evidence | Contextual factors influencing intervention/ evaluation[3] | No. of domains measured | No. of outcomes measured | Quality rating |
|---|---|---|---|---|---|---|---|---|---|---|---|---|---|---|---|
| Belsky, 2006 [41] Melhuish, 2007 [43] Melhuish, 2008 [42] Melhuish, 2010 [8] National Evaluation of Sure Start Team, 2010 [44] Melhuish, 2011 [46] National Evaluation of Sure Start Team, 2012 [45] | **Sure Start (UK)** | Initially implemented in 248 (scaled up to 500) disadvantaged locations across the UK from 1998 and aimed to improve existing universal services and plug identified service gaps. | Small areas with average populations of just under 13000 people including about 700 children aged 0–3 years. | SSLPs were targeted to 20% of the most deprived areas in England | **Reshaping, enhancing existing services and increasing coordination between services** will lead to improvement **in the services that are delivered**, and result in enhanced child, family and community functioning. | 0–4 | Ongoing (reconfigured into Children's Centres) | 7 years | 1–3 | Initial study 2, 3, 7a  Subsequent design 2, 4, 6, 7a[4] | III-2 | 2, 3 | 5 | 19 | Medium |
| NNI Research Team, 2007 [47] | **Neighbourhood Nurseries Initiative (UK)** | Aimed to create 45,000 new high quality childcare places in disadvantaged areas. Commenced in 2001. Early education and family support was also to be provided by centres. | Targeted nursery 'places' in disadvantaged neighbourhoods. | Neighbourhood Nurseries were targeted to disadvantaged neighbourhoods as defined by the Index of Multiple Deprivation, but local authorities could make a case for the location of a nursery outside of these areas if there was evidence of pockets of deprivation in more affluent areas.  Nurseries were to be located new major roads on a 'travel to work' basis so that higher income parents would take up non-NNI funded places for social mix and sustainability.  For the purposes of evaluation, areas were grouped into 'low' and 'high' NNI resource areas | **Funding additional childcare places** in the most disadvantaged neighbourhoods in the country **will enable disadvantaged parents to work,** which would in turn provide better conditions, opportunities and outcomes for children. Ensuring children had **access to high quality childcare** will also improve outcomes for children. | 0–4 | Time limited (absorbed into Sure Start) | 4 years | 1–3 | Tracking study: 2, 3, 5, 7a  Impact study on families (3 different approaches):1) 2, 3, 7a  2) 2, 3, 7b  3) 2, 3, 8 | III-2 | 1, 3, 4 | 2 | 2 | Medium |

*(Continued)*

**Table 3.** (Continued)

| First author(s), Year | Initiative | Description | Size of local delivery area | Spatial targeting | Theory of change | Target age of children | Ongoing or time-limited | Time between implementation & last outcomes evaluation | Overall evaluation designs[1] | Impact study design[2] | Level of evidence | Contextual factors influencing intervention/evaluation[3] | No. of domains measured | No. of outcomes measured | Quality rating |
|---|---|---|---|---|---|---|---|---|---|---|---|---|---|---|---|
| White, 2010 [48]; Knibbs, 2013 [49]; Heaven, 2014 [50]; Wilton, 2017 [51] | **Flying Start (Wales)** | Implemented in disadvantaged communities in Wales (number unclear) from 2006. Provided 4 key entitlements: enhanced health visiting, parenting support, support for early language development, free high quality childcare. | School catchment areas | Deprived local areas were initially chosen and target catchments in these areas were then further specified. There was a 16,000 cap on the number of children who could participate so the Flying Start areas are highly concentrated and cover a small proportion of relevant Local Authorities. School catchment areas were chosen as they were thought to be understood by parents, provide for clear definitions of the target areas, enabled a community focus and links to be established with other services, and facilitated measurement when they moved up to school. | **Intensively provide four core services universally** to designated flying start areas, **with some additional discretionary support according to identified local need**, to achieve medium term improvements in outcomes for children and their families, and a long term decisive difference to the life chances of children under 4. | 0–4 | Ongoing | 9 years | 1–3 | 2, 4, 6, 7a | III-2 | 7 | 4 | 14 | High |

**State /Regional Initiatives**

(Continued)

**Table 3.** (Continued)

| First author(s), Year | Initiative | Description | Size of local delivery area | Spatial targeting | Theory of change | Target age of children | Ongoing or time-limited | Time between implementation & last outcomes evaluation | Overall evaluation designs[1] | Impact study design[2] | Level of evidence | Contextual factors influencing intervention/ evaluation[3] | No. of domains measured | No. of outcomes measured | Quality rating |
|---|---|---|---|---|---|---|---|---|---|---|---|---|---|---|---|
| Raban, 2006 [52] Kelaher, 2009 [53] Kelaher, 2009 [54] | **Best Start (Victoria, Australia)**[5] | Initially implemented in 11 disadvantage locations in Victoria, Australia from 2002. Designed to improve health, development, learning and wellbeing of children by increasing cooperation, collaboration and coordination between existing universal services. | Varied. Some were whole municipalities, some were rural areas, some were a collection of small towns, which didn't fit neatly into defined Statistical Local Areas or Local Government boundaries. | A mix of demonstration sites were chosen across metropolitan, regional and rural areas. No information is given about reasons for the specific sites that were chosen except that two sites were specifically chosen to focus on Aboriginal communities. The sites fell into 4 categories: whole municipalities, building on other initiatives, rural/small towns projects, Aboriginal projects. | Rather than introducing new services or expand existing services, Best Start will **increase cooperation, collaboration and coordination between universal early years services** so they are more responsive to local needs. Best Start will **support services to move across traditional boundaries,** using **active community involvement.** This will lead to improved outcomes for children, and in particular improved access for vulnerable families. | Pregnancy to early school years (age not specified) | Ongoing | 2 years | 1, 2 | 2, 3, 5, 7b, 8 Range of study designs used. Main outcomes data: 11 sites compared to 1) historical control and 2) rest of state | III-2/ III-3 | 7 | 4 | 10 | Medium |
| Browning, 2010 [55] Compass Evaluation & Research, 2015 [56] | **First Steps**[5] **(South Carolina, USA)** | Implemented in all counties in South Carolina as its families were among the most disadvantaged in the country from 1999. The primary aim was school readiness, with a focus on health, early learning and mobilising communities. Improving quality of universal services such as childcare and kindergarten became an increasing focus as the initiative progressed. | County | South Carolina is one of the most disadvantaged states in the USA. It is the 10th poorest state in the USA; 48% of children live in low-income families and was ranked 45/50 by Kids Count for child wellbeing. The decision was made by the governor to provide First Steps to all counties in South Carolina | State and **local partnerships will select and implement strategies to respond to identified needs.** These strategies will result in specific outcomes, and outcome achievement will result in children whose development is optimized and are ready to start school. | 0–5 | Ongoing | 6 years | 1, 2 | 2, 3, 6, 8 | III-2 | 1, 2, 3, 4, 5 | 1 | 3 | Medium |

(Continued)

**Table 3.** (Continued)

| First author(s), Year | Initiative | Description | Size of local delivery area | Spatial targeting | Theory of change | Target age of children | Ongoing or time-limited | Time between implementation & last outcomes evaluation | Overall evaluation designs[1] | Impact study design[2] | Level of evidence | Contextual factors influencing intervention/ evaluation[3] | No. of domains measured | No. of outcomes measured | Quality rating |
|---|---|---|---|---|---|---|---|---|---|---|---|---|---|---|---|
| Bryant, 2004 [58] Ladd, 2014 [57] | **Smart Start (North Carolina, USA)** | Implemented in all counties in North Carolina from 1993. It aimed to improve the delivery of services to all children <5 by responding to disadvantages that many children experience. It had 3 program areas: childcare quality, family functioning and child health. Interagency collaboration was a means by which this could be achieved. | County | Smart Start started as a demonstration project in 18 counties and was gradually expanded to all counties across the state. | **Strengthen the child care system, support family functioning** and **access to child health care, and improve interagency collaboration** to prepare children for school. | 0–5 | Ongoing | 16 years | 1, 2 | 4, 6, 7a | III-3 | 1, 3, 5 | 1 | 1 | Medium |
| Darnell, 2013 [59] | **Georgia Family Connection (Georgia, USA)** | Implemented in all counties in Georgia. Commenced in 1990. It created networks of community collaborates with 5 key focus areas: healthy children, children ready to start school, children succeeding at school, stable, self-sufficient families and strong communities. This evaluation only examined the prevention of low birth weight. | County (range 1.6k – 1.05m) | In the inaugural national Kids Count report released in 1990 Georgia placed 48th out of 50 states. The Governor funded a two-year demonstration project in 15 counties. Over 10 years the State gradually increased funding and made it a Statewide initiative and there is now a Community Collaborative in every County. It was part of a long-term commitment to improve the well-being of children and families. | Collaborative operations will affect birthweight through **individual and community level pathways,** based on **identified needs in each community.** The theory of change **is not prescriptive about any particular model of intervention.** | Pregnancy –4 | Ongoing | Concurrent/ mixed | - | 2, 3, 9, 7b | III-2 | 6 | 1 | 1 | Medium |

**Demonstration Projects**

*(Continued)*

**Table 3.** (Continued)

| First author(s), Year | Initiative | Description | Size of local delivery area | Spatial targeting | Theory of change | Target age of children | Ongoing or time-limited | Time between implementation & last outcomes evaluation | Overall evaluation designs[1] | Impact study design[2] | Level of evidence | Contextual factors influencing intervention/ evaluation[3] | No. of domains measured | No. of outcomes measured | Quality rating |
|---|---|---|---|---|---|---|---|---|---|---|---|---|---|---|---|
| Corter, 2007 [60] Corter, 2008 [62] Corter, 2009 [61] Corter, 2012 [63] | **Toronto First Duty (Toronto, Canada)** | Implemented in 5 areas in Toronto from 2001. It aimed to integrate universal services with other family support services in hubs in primary schools in order to improve equity of access and outcomes for children. | Neighbourhoods based around schools | Each school site was chosen for different reasons–eg one was selected due to its dense population and cultural and linguistic diversity, another was chosen due to the threat of school closure, while another was chosen as it already had a strong history of successful collaboration. | **Colocation and service integration** of **fragmented universal services will** result in improved program quality and outreach to the underserved and improve outcomes for children, their families and communities | 0–6 | Time limited | 4 years | 1–3 | Two designs used: 1) 3, 5, 7a 2) 2, 7a | 1) III-3 2) III-2 | 1, 3, 4 | 1 | 4 | Low |
| McKeown, 2014 [64] | **National Early Years Access Initiative (NEYAI) (Ireland)** | 4 year demonstration program that commenced in 2011 in 11 metropolitan and 2 rural disadvantaged areas in Ireland that aimed to improve quality and outcomes in the early years sector. It included a focus on improving the quality of the free pre-school year. | Suburbs or collection of suburbs in urban areas in Dublin, Cork and Limerick and two rural areas | It is not specified why these areas were targeted. | **Evidence-based programs** delivered to children and parents, as well as **training and mentoring of staff** will deliver improved staff capacity and improved child outcomes. | 0–6 | Time limited | 2.5 years | 2 | 2, 4, 5, 8 | III-2 | 7 | 1 | 4 | Low |

*(Continued)*

**Table 3.** (Continued)

| First author(s), Year | Initiative | Description | Size of local delivery area | Spatial targeting | Theory of change | Target age of children | Ongoing or time-limited | Time between implementation & last outcomes evaluation | Overall evaluation designs[1] | Impact study design[2] | Level of evidence | Contextual factors influencing intervention/ evaluation[3] | No. of domains measured | No. of outcomes measured | Quality rating |
|---|---|---|---|---|---|---|---|---|---|---|---|---|---|---|---|
| Mackenzie, 2004 [66] Shute, 2005 [65] | **Starting Well (Glasgow, Scotland)** | Implemented in 2 disadvantaged areas in Glasgow from 2000. Provided an enhanced home visiting service, improved community support, and the development of integrated organisational services. | A collection of suburbs in two areas in Glasgow | The two geographical areas within the City of Glasgow were chosen due to their relative socio-economic disadvantage. | **Augmented home visiting,** combined with **integrated, enhanced local community supports and structures** will reduce child morbidity and increase the number of healthy, functioning families | Young children and their families (age not specified) | Time limited | 2 years | 1 | 2, 4, 6, 8 | III-2 | 7 | 3 | 4 | Low |

1 Evaluation Design Codes (in addition to Impact Study): 1 Process, 2 Local evaluation, 3 Economic/Cost-effectiveness

2 Impact Study Design Codes: 1 RCT 2 Quasi-experimental 3 Cross-sectional 4 Cohort 5 Pre- & post- 6 Longitudinal 7a) Population sample–general 7b) Population sample–intervention areas 8 Intervention sample 9 Time series

3 Factors influencing intervention/evaluation codes: 1 Intervention funding changes 2 Intervention scope changes 3 Intervention design changes 4 Broader Policy impacts on population 5 Evaluation funding/scope changes 6 Unknown/unclear 7 None

4 Subsequent design compared Sure Start intervention sites with children/families from Millennium Cohort Study

5 Evaluation of participants attending specific programs (such as home visiting) are not included here as these focus on pre- and post- change of those enrolled in these programs only.

[60]. The ABC Programme selected bounded areas in which resident populations identified with each other as a community [40]. Neighbourhood Nurseries Initiatives aimed to increase nursery 'places' in disadvantaged neighbourhoods, and expected any new nurseries be located near major roads [47]. Communities for Children sites were chosen based on criteria for multiple aspects of disadvantage and each site was defined differently, from a collection of postcodes to one or more defined Statistical Local Areas [37]. Similarly, Best Start sites ranged from whole municipalities to a small collection of rural towns or areas with a high Aboriginal population [52]. In the smaller demonstration projects Starting Well and NEYAI, the target delivery areas were described as a collection of suburbs [64, 66].

**Theories of change.** A theory of change (or program logic model) explains how and why an initiative is intended to work [68]. From an evaluation perspective, the value of articulating a theory of change for complex initiatives is that it helps evaluators understand not just whether and how an initiative works, but which parts of an initiative have the greatest impact on outcomes [68]. We appraised all included initiatives to determine whether a theory of change had been developed. We found all initiatives had articulated a theory of change, either in text or figure form, as summarised in Table 3. All but one initiative (Neighbourhood Nurseries Initiative) had collaboration/partnership as a component of their theory of change, with this considered a 'key ingredient' to success for many. For example, Georgia Family Connection [59] theorised that its collaboration model was the primary difference between it and the comparison group. All but one initiative (Communities for Children) included modified universal services as part of their logic model, with three initiatives (Georgia Family Connection [59], First Steps [55, 56], Starting Well [65, 66]) also including the development of additional targeted services in their model. Communities for Children [38] theorised that plugging unmet service gaps would improve outcomes. Ten initiatives (Communities for Children [38], ABC Programme [40], Sure Start [8], Flying Start [48], Best Start [52], First Steps [55, 56], Smart Start [58, 67], Georgia Family Connection [59], NEYAI [64], Starting Well [65, 66]) theorised that involving the local community in decision-making would be beneficial; and all twelve initiatives included some degree of local area autonomy in their model.

## Evaluation designs

Given the complexity of public place-based initiatives, evaluations may contain multiple elements, including: process evaluation, local evaluations, an economic or cost effectiveness evaluation, and an impact evaluation. We assessed the evaluation designs of each initiative according to these elements. First we applied the quality ratings (Table 3 and S2 Appendix); then we assessed whether the various components of evaluation were undertaken in addition to an impact study. Finally, we looked at design and methods used for impact studies. These are briefly defined and then discussed in each of the sub-sections below.

**Quality.** The evaluations of two initiatives were classified as high quality (Communities for Children [10, 37–39], Flying Start [48–51]), six as medium quality (Sure Start [8, 41–46], Neighbourhood Nurseries Initiative [47], Best Start [52–54], First Steps [55, 56], Smart Start [58, 67], Georgia Family Connection [59]), and four as low quality (ABC Programme [40], Toronto First Duty [60–63], NEYAI [64], Starting Well [65, 66]) (Table 3 and S2 Appendix).

**Evaluation design overview.** Five initiatives (Sure Start [8, 41–46], Neighbourhood Nurseries Initiative [47], Flying Start [48–51], Communities for Children [10, 37–39], Toronto First Duty [60–63]) had a comprehensive evaluation design that combined the impact evaluation with process evaluation, local evaluation, and/or some cost-benefit or cost-effectiveness analysis. Comprehensive designs were a particular feature of the large national initiatives in the UK and Australia. Within these broad elements, evaluation designs took a range of forms. For the

large, national initiatives like Sure Start [8], Communities for Children [37, 38] and Flying Start [48, 49], evaluation designs aligned with the structure outlined in Fig 1. Some initiatives applied a specific evaluation model to their evaluation (Best Start [52]), while others used more generic evaluation terms to describe their evaluation approach, e.g., 'formative' and 'summative' (Toronto First Duty [60]).

For all initiatives, the evaluation was commissioned to independent external evaluators. Nine appeared to have their evaluations commissioned and designed after implementation had commenced resulting in a lack of pre-intervention baseline data (Flying Start [49], Starting Well [65, 66]), delays in the commencement of data collection (Flying Start [49]) and the use of less-than-ideal datasets. An example of this is the NEYAI evaluation, which was based on children who participated in a year of free pre-school and received the NEYAI intervention, and compared them to children who attended another type of free pre-school [64]. The evaluation report focussed more on the benefits of pre-school than on the benefits of NEYAI. Two initiatives received funding for an impact evaluation a long time after the initiative had been implemented (Georgia Family Connection [59], Smart Start [67]). For example, evaluation funding for Smart Start ceased after 10 years [58] without a whole initiative evaluation having been conducted. Philanthropic funding was made available some years later to evaluate longer term outcomes of the program using routinely collected data [57].

**Process evaluation.**   Process evaluation seeks to understand the explanatory elements that may influence the outcome of an intervention [69]. It helps to determine whether an intervention's failure to show any positive effects is due to the design of intervention itself or due to poor implementation [69]. Traditional process evaluation includes an assessment of quality, reach, dosage, satisfaction and fidelity [70]. For place-based initiatives, additional process evaluation considerations may include how to measure whether organisations are working in a 'joined-up' way and the level of community involvement in decision-making, if these were part of the theory of change [6]. None of the initiatives comprehensively evaluated all the expected elements of process evaluation with a whole-of-initiative synthesis. There was considerable diversity in the approaches that were taken to process evaluation, although some commonalities were apparent.

Of the ten process evaluations that were conducted (Communities for Children [71], ABC Programme [40], Sure Start [43], Neighbourhood Nurseries Initiative [47], Flying Start [48], Best Start [52], First Steps [55, 56], Smart Start [58], Toronto First Duty [60, 61, 63], Starting Well [66]), there was broad alignment between the aims of the initiatives and the process evaluation designs. For example, initiatives that aimed to improve service quality strongly focussed on measuring service quality indicators such as kindergarten or childcare quality (First Steps [55, 56], Neighbourhood Nurseries Initiative [47]), while initiatives that aimed to improve access to services measured reach (Communities for Children [71], First Steps [55, 56], Neighbourhood Nurseries Initiative [47]). Two initiatives that had a specific focus on joined-up working and partnerships as a means for improving service coordination, conducted assessments of the difference in this pre- and post-implementation (Communities for Children [71], Best Start [52]). Initiatives that aimed to build service capacity developed service profiles and looked at the difference in the number of services available pre- and post- (Communities for Children [71], Neighbourhood Nurseries Initiative [47]). The ABC Programme [40] was the only initiative to include a specific aim to increase the use of evidence and data in decision-making, and their process evaluation assessed reported changes in the use of evidence and data in local planning and service delivery. Other features typical of process evaluation designs included the collection of 'performance monitoring indicators', and number and type of services provided.

Fidelity was not commonly examined by the initiatives. First Steps was a notable exception, and undertook an examination of fidelity of their programs against pre-defined Program Accountability Standards [56]. They found an improvement in the fidelity of implementation over a two-year period, with a particularly high degree of fidelity for mature evidence-based programs.

Sure Start's process evaluation framework was comprehensive and the findings span multiple reports, not all of which could be included in this review. A key finding was that due to the rapid scale-up of the program, and the variation in the number and type of programs being implemented, the quality of programs being delivered varied widely [8]. Moreover, they found a relationship between well implemented programs and better outcomes for children [43].

**Local evaluation.**   Local evaluation is where each geographic area (e.g., community or neighbourhood) evaluates its own activity. Collecting and synthesising local evaluation learnings provides valuable explanatory evidence about how and why initiatives may or may not be working as intended. Previous research has highlighted the challenges in collecting local evaluative data in a format that is both meaningful for local management and that enables whole-of-initiative synthesis [16, 17]. We identified and briefly appraised any findings that were collated in whole-of-initiative evaluation studies. Eight initiatives included local evaluation as part of their evaluation design (Communities for Children [71], Sure Start [8], Neighbourhood Nurseries Initiative [47], Flying Start [48], Best Start [52], First Steps [56], Smart Start [58], Toronto First Duty [60, 61, 63], NEYAI [64]). These primarily examined process elements that took into account the local geographic context. Evaluators noted that local variation in existing infrastructure, community capacity, networks and rurality impacted on implementation. Others observed that arbitrary administrative boundaries conflicted with the local place boundaries set by the initiative.

**Impact study designs.**   Impact (or outcome) evaluations examine the positive and negative effects of an intervention, using a set of pre-specified outcome measures [72]. An inclusion criteria for this review was that an impact study had been conducted. We examined the design of each impact study, the dataset(s) used, length of study, and the number and range of outcomes assessed (Table 1). Table 3 contains an overview of the findings for each initiative.

Impact evaluation studies varied considerably in design. Some initiatives used a combination of designs and data sources to assess impact. The ABC Programme [40] is described last in the following summary, as it was the only initiative that did not include a quasi-experimental design in their evaluation.

For the quasi-experimental impact evaluations, broadly, three types of sampling approaches were employed. Six initiatives (Communities for Children [10, 39], Sure Start [41, 42, 44, 45], Neighbourhood Nurseries Initiative [47], Flying Start [49–51], Smart Start [67], Toronto First Duty [62]) used a general population sample from geographic areas where the initiative was conducted, irrespective of which elements of the possible initiative had been delivered and irrespective of whether or not the sample had actually received any form of intervention. This approach sought to determine the whole-of-community, population level impact of the initiative. In a more tailored approach, three initiatives (Best Start [52–54], Georgia Family Connection [59], Neighbourhood Nurseries Initiative [47]) used an 'intervention area' or 'targeted' population sample. Again population level data were examined, but only included geographic areas where it was known that interventions designed to improve specific outcomes of interest had been implemented (for example, in Best Start, examination of breastfeeding rates only in the communities where a breastfeeding program had been provided [53]). Five initiatives (Neighbourhood Nurseries Initiative [47], Best Start [52], First Steps [55], NEYAI [64], Starting Well [65, 66]) assessed individual-level impact, using the less optimal approach of intervention samples comprising only participants known to have received some form of the

intervention. Several initiatives used more than one type of design, using population-level data where available, and supplementing this with individual-level data for some outcomes of interest.

Seven initiatives used the stronger design of a cohort sample (Communities for Children [10, 39], Sure Start, Flying Start [49–51], Smart Start [67], NEYAI [64], Starting Well [65, 66]), while six used a cross-sectional sample (Sure Start [41], Neighbourhood Nurseries Initiative [47], Best Start [52–54], First Steps [55], Georgia Family Connection, Toronto First Duty [62]). Sure Start used both, reflecting a change in their study design part-way through the evaluation. Two initiatives used only their own collected data to assess impact (Communities for Children [10, 39], NEYAI [64]), four used only secondary datasets (Smart Start [67], Georgia Family Connection [59], Toronto First Duty [62], Starting Well [65, 66]), while five used a mix of both (Sure Start [41, 42, 44, 45], Flying Start [49–51], Best Start [52–54], Neighbourhood Nurseries Initiative [47], First Steps [55]). Initiatives using secondary datasets were more likely to have a cross-sectional impact study design.

The ABC Programme [40] used a pre- and post- evaluation design, comparing outcomes for parents and children who participated in the initiative (i.e., intervention sample). The initiative collected its own data using a set of core measures.

Four initiatives (ABC Programme [40], Best Start [52], NEYAI [64], Starting Well [66]) were most recently evaluated within three years of implementation. This was more common in demonstration projects. The longest time participants were followed up after implementation ranged between two years and 16 years, with a four to five year timeframe being the most common.

## Contexts in which initiatives were implemented and evaluated

The context in which initiatives are implemented and evaluated can affect their results [69]. We examined the evaluation reports for each initiative to assess them for reported changes in funding, scope, design and broader policy contextual changes which may have impacted on outcomes. Many of the initiatives and their evaluations were subject to such changes. Four initiatives reported a fluctuation or reduction in funding during the life of the initiative. Funding cuts were reported due to government austerity measures in response to the Global Financial Crisis (First Steps [55]) or a change in government (Toronto First Duty [60]). Two initiatives noted changes but were silent on the reason (Communities for Children [39], Smart Start [67]). In addition, three (Communities for Children [39], First Steps [55], and Smart Start [58]) reported a reduction in funding for evaluation which reduced the planned scope, and in one case (Smart Start) led to a temporary cessation of evaluation activities.

Three initiatives (Communities for Children [39], Sure Start [8], First Steps [55]) reported a change in scope. For example, Communities for Children increased the age of targeted children from 0–5 to 0–12 without any increase in funding. Six initiatives reported a change in design, including being subject to a greater level of 'top-down' prescription. The transformation from Sure Start's 'Local Programmes' to 'Children's Centres' resulted in services and guidelines being more clearly specified [8]. The second evaluation of First Steps recommended that the initiative should prioritise funding for early education and childcare over parenting programs and family literacy [55]. Smart Start increased the required total percentage of funds to be spent on childcare related activities from 30 percent to 70 percent [67]. Three studies encouraged or mandated the use of evidence-based programs (Sure Start [8], Communities for Children [39], First Steps [56]).

Four initiatives (Communities for Children [39], Neighbourhood Nurseries Initiative [47], First Steps [55], Toronto First Duty [60]), discussed broader policy changes at a national and

state level which impacted the initiatives. For example, the Neighbourhood Nurseries Initiative was gradually absorbed into Sure Start while the evaluation was occurring, and in Canada a change of government altered the way childcare was funded and directly affected the Toronto First Duty model and the families accessing its services.

## Outcomes–are place-based initiatives effective?

Outcome domains were summarised into five categories: pregnancy and birth, child, parent, family, and school and community. A summary of the findings for each initiative is provided in Table 4. Detailed tables are available in S3 Appendix. Outcomes in the pregnancy and birth category were the least commonly evaluated while those in the child category were most commonly examined. The initiatives evaluated between one and 19 outcome domains each, with a total of 88 outcomes measured across the 12 initiatives. Despite having broadly-based goals and objectives, two initiatives (Georgia Family Connections [59] and Smart Start [57]) were evaluated using only one outcome each. The 11 initiatives with a comparison group will be discussed first (Communities for Children [10, 38, 39], Sure Start [41, 42, 44, 45], Neighbourhood Nurseries Initiative [47], Flying Start [49–51], Best Start [52–54], First Steps [55], Smart Start [57], Georgia Family Connection [59], Toronto First Duty [62], NEYAI [64], Starting Well [65, 66]), followed by the ABC Programme [40], whose non-experimental design necessitates separate consideration.

For all 11 initiatives with a comparison group, evidence of effectiveness was mixed across all domains. Across the 83 outcome domains reported, 30 (36.4%) demonstrated a positive outcome, and all but one initiative (NEYAI [64]) demonstrated a positive outcome in at least one outcome measure. Of the studies that examined outcomes more than once post baseline (Communities for Children [39], Sure Start [44, 45], First Steps [55], Smart Start [57], Georgia Family Connection [59], and Starting Well [66]), 10 from 38 outcomes (26.3%) demonstrated positive sustained results.

The child domain had the lowest proportion of reported positive effects (8 of 31 measured, 25.8%). Of the seven outcomes measured more than once, two (28.6%) found sustained positive results. Positive results were more likely to be seen in the school and community domain, in 10 of 16 outcomes measured (62.5%), with three from nine (33%) showing a sustained positive result when measured more than once. This is followed by pregnancy and birth (55.5%), with the one outcome measured more than once showing sustained positive results. The parent domain had 41.6% of outcomes measured demonstrating a positive result, with only one from nine (11.1%) showing a sustained positive resulted when measured more than once. Finally, the family domain had five from 15 outcomes demonstrating a positive result (33.3%), with three from 10 (30%) showing a sustained positive result. Adverse effects were found in four outcomes measured: one in the child domain, two in the parent domain, and one in the school and community domain.

The non-experimental ABC Programme [40] measured three child domain outcomes and two family domain outcomes, and demonstrated a positive result for all five outcomes.

**Synthesis of results.** Table 5 draws together information about the design of initiatives, their impact study design, theories of change and positive pregnancy & birth/child outcomes at population level to assist in drawing conclusions about effectiveness. It is difficult to draw definitive conclusions given the mixed quality, with three studies that did not measure outcomes at the population level, only four studies that measured whether outcomes were sustained over time, and one study that used a non-experimental design. Nevertheless, some inferences can be made. For the eight initiatives that used a population level sample, all found evidence of impact. For the four initiatives that measured population level impact over time

**Table 4. Study reported outcomes–summary by category.**

| First author(s), year | Initiative | Impact study design | Pregnancy & birth | Child | Parent | Family | School & community | Total |
|---|---|---|---|---|---|---|---|---|
| **Studies with comparison group** | | | | | | | | |
| Edwards, 2009 [38] | **Communities for Children (Australia)** | General population sample, cohort design | - | 1 −ve effect, not sustained | 1 +ve effect, not sustained | 2 +ve effect, 1 sustained | 1 +ve effect, sustained | **15 measured** |
| Edwards, 2011 [10] | | | | 2 no/weak effect | 1 −ve, sustained | 3 no/weak effect | 3 no/weak effect | **4 +ve effect, 2 sustained** |
| Edwards 2014 [39] | | | | | 1 no/weak effect | | | **2 −ve effect, 1 sustained** |
| | | | | | | | | **9 no/weak effect** |
| Belsky, 2006 [41] | **Sure Start (UK)** | General population sample, cohort design | 2 no/weak effect | 3 +ve effect, 1 sustained | 2 +ve effect, 1 sustained | 3 +ve effect, 2 sustained | 1 +ve effect, not sustained | **19 measured** |
| Melhuish, 2008 [42] | | | | 2 no/weak effect | 1 −ve effect, not sustained | 1 no/weak effect | 1 −ve effect, not sustained | **9 +ve effect, 4 sustained** |
| National Evaluation of Sure Start Team, 2010 [44] | | | | | 2 no/weak effect | | 1 no/weak effect | **2 −ve effect** |
| National Evaluation of Sure Start Team, 2012 [45] | | | | | | | | **8 no/weak effect** |
| NNI Research Team [47] | **Neighbourhood Nurseries Initiative (UK)** | General & targeted population samples (2 studies), cross-sectional design | - | 1 no/weak effect | 1 +ve effect | - | - | **2 measured** |
| | | | | | | | | **1 +ve effect** |
| | | | | | | | | **1 no/weak effect** |
| Knibbs, 2013 [49] | **Flying Start (Wales)** | General population sample, cohort design | 2 +ve effect | 3 +ve effect | 2 no/weak effect | 4 no/weak effect | 4 +ve effect | **20 measured** |
| Heaven, 2014 [50] | | | 2 no/weak effect | 3 no/weak effect | | | | **9 +ve effect** |
| Wilton, 2017 [51] | | | | | | | | **11 no/weak effect** |
| Raban, 2006 [52] | **Best Start (Australia)** | Targeted population sample, cross-sectional design | 1 +ve effect | 4 no/weak effect | - | 1 no/weak effect | 2 +ve effect | **10 measured** |
| Kelaher, 2009 [53] | | | 1 no/weak effect | | | | 1 no/weak effect | **3 +ve effect** |
| Kelaher, 2009 [54] | | | | | | | | **7 no/weak effect** |
| Browning, 2010 [55] | **First Steps (USA)** | Intervention sample, cross-sectional design | - | 3 mixed effects | - | - | - | **3 measured** |
| | | | | | | | | **3 mixed effect** |
| Ladd, 2014 [57] | **Smart Start (USA)** | General population sample, cohort design | - | 1 +ve effect, sustained | - | - | - | **1 measured** |
| | | | | | | | | **1 +ve effect, sustained** |
| Darnell, 2013 [59] | **Georgia Family Connection (USA)** | Targeted population sample, cross-sectional design | 1 +ve effect, sustained | - | - | - | - | **1 measured** |
| | | | | | | | | **1 +ve effect, sustained** |
| Corter, 2008 [62] | **Toronto First Duty (Canada)** | General population sample, cross-sectional design | - | 1 +ve | - | - | - | **4 measured** |
| | | | | 3 no/weak effect | | | | **1 +ve effect** |
| | | | | | | | | **3 no/weak effect** |
| McKeown, 2014 [64] | **NEYAI (Ireland)** | Intervention sample, pre-/post- design | - | 4 no/weak effect | - | - | - | **4 measured** |
| | | | | | | | | **4 no/weak effect** |

(*Continued*)

**Table 4.** (Continued)

| First author(s), year | Initiative | Impact study design | Pregnancy & birth | Child | Parent | Family | School & community | Total |
|---|---|---|---|---|---|---|---|---|
| Mackenzie, 2004 [66] | **Starting Well (Scotland)** | Intervention sample, cohort design | - | - | 1 +ve effect, not sustained | 1 no/weak effect | 2 +ve effect, sustained | **4 measured** |
| Shute, 2005 [65] | | | | | | | | **3 +ve effect, 2 sustained** |
| | | | | | | | | **1 no/weak effect** |
| | **Total** | | **9 measured** | **31 measured** | **12 measured** | **15 measured** | **16 measured** | **83 measured** |
| | | | **2+ve effect, 1 sustained** | **8 +ve effect, 2 sustained** | **5 +ve effect, 1 sustained** | **5 +ve effect, 3 sustained** | **10 +ve effect, 3 sustained** | **30 +ve effect, 10 sustained** |
| | | | **3 no/weak effect** | **1 −ve effect** | **2 −ve effect, 1 sustained** | **9 no/weak effect** | **1 −ve effect** | **4 −ve effect, 1 sustained** |
| | | | | **19 no/weak effect** | **5 no/weak effect** | | **5 no/weak effect** | **41 no/weak effect** |
| | | | | **3 mixed effects** | | | | **3 mixed effects** |
| **Studies with no comparison group** | | | | | | | | |
| Hickey, 2018 [40] | **ABC (Ireland)** [40] | Intervention sample, pre-/post- design | | **3 measured** | | **2 measured** | | **5 measured** |
| | | | | **3 +ve effect** | | **2 +ve effect** | | **5 +ve effect** |

NOTE: +ve indicates positive or−ve indicates negative effect at P≤.05; If measured more than once, sustained effect is indicated.

(best design), three found evidence of sustained impact, but for one measure only. Given place-based initiatives are expected to improve outcomes across a range of measures, this is a somewhat disappointing result. Initiatives that used a targeted population sample were most likely to report positive results. For example, Best Start only measured the impact of the initiative on breastfeeding rates with communities where it was known that breastfeeding was specifically targeted, and found a positive effect [53]. Similarly, Georgia Family Connection identified the communities that targeted low birth weight and only included these communities in their study design. They too found a positive effect [59]. Initiatives that used routinely collected datasets to measure outcomes over longer time periods (Georgia Family Connection [59], Smart Start [57]) were more likely to demonstrate positive outcomes compared to purposely designed studies, yet were able to measure fewer outcomes due to the limitations of data availability. Initiatives that used a general population sample and a purposely designed study sample for their impact study and used a broader range of measures were less likely to find sustained positive effects (Communities for Children, Sure Start), although Communities for Children and Sure Start found positive effects in the early years that were not sustained over time [39, 45]. The ABC Programme [40] found positive effects across all outcomes it measured, however its pre- and post- evaluation design is considered a lower level of evidence compared to the more robust quasi-experimental design employed by the other initiatives examined.

Some initiatives used multiple designs within their evaluation framework. For example, the Neighbourhood Nurseries Initiative [47] used three different samples to assess for impact. In a general population sample (all parents living in a Neighbourhood Nursery Initiative 'rich' area) there was no evidence of impact on work status and childcare uptake. Similarly, in a targeted population sample (parents who were identified as being 'work ready' and living in a Neighbourhood Nursery Initiative 'rich' area) there was no evidence of impact. However in an

**Table 5. Synthesis of results.**

| First author(s), year | Initiative (quality rating) | Size of local delivery areas | Sample data for impact evaluation (study length) | Mechanisms by which child and family outcomes will be achieved | | | | | Evidence of positive impact on *pregnancy/birth/child* outcomes at population level | | |
|---|---|---|---|---|---|---|---|---|---|---|---|
| | | | | Enhance, intensify, collocate or redesign universal services | Address unmet service gaps | Joined up working / collaboration | Community involvement | Local discretion /variation | Measured at population level? General (G) or Targeted (T) sample | Evidence of impact at population level? Cohort (CO) or Cross-sectional (CS) | Evidence of sustained impact? |
| **Studies with comparison group** | | | | | | | | | | | |
| Katz, 2007 [37]<br>Edwards, 2009 [38]<br>Edwards, 2011 [10]<br>Edwards 2014 [39] | **Communities for Children** (high) | Variable and variably defined | Study designed and collected (5 years) | | √ | √ | √ | √ | √ (G) | √ (CO) | X |
| Belsky, 2006 [41]<br>Melhuish, 2007 [43]<br>Melhuish, 2008 [42]<br>Melhuish, 2010 [8]<br>National Evaluation of Sure Start Team, 2010 [44]<br>Melhush, 2011 [46]<br>National Evaluation of Sure Start Team, 2012 [45] | **Sure Start** (medium) | Small and variably defined | Study designed and collected + secondary dataset (7 years) | √ | | √ | √ | √ | √ (G) | √ (CO) | √ (1 measure only) |
| Bryant, 2004 [58]<br>Ladd, 2014 [57] | **Smart Start** (medium) | Large and uniformly defined | Secondary datasets (16 years) | √ | | √ | √ | √ | √ (G) | √ (CO) | √ (1 measure only) |
| Darnell, 2013 [59] | **Georgia Family Connection** (medium) | Large and uniformly defined | Secondary datasets (mixed) | √ | √ | √ | √ | √ | √ (T) | √ (CS) | √ (1 measure only) |
| White, 2010 [48]<br>Knibbs, 2013 [49]<br>Heaven, 2014 [50]<br>Wilton, 2017 [51] | **Flying Start** (high) | Small and well defined | Study designed and collected + secondary dataset (9 years) | √ | | √ | √ | √ | √ (G) | √ (CO) | N/A |

(*Continued*)

**Table 5.** (*Continued*)

| First author(s), year | Initiative (quality rating) | Size of local delivery areas | Sample data for impact evaluation (study length) | Mechanisms by which child and family outcomes will be achieved | | | | | Evidence of positive impact on *pregnancy/birth/child* outcomes at population level | | |
|---|---|---|---|---|---|---|---|---|---|---|---|
| | | | | Enhance, intensify, collocate or redesign universal services | Address unmet service gaps | Joined up working / collaboration | Community involvement | Local discretion /variation | Measured at population level? General (G) or Targeted (T) sample | Evidence of impact at population level? Cohort (CO) or Cross-sectional (CS) | Evidence of sustained impact? |
| Raban, 2006 [52] Kelaher, 2009 [53] Kelaher, 2009 [54] | **Best Start** (medium) | Variable and variably defined | Study designed and collected + secondary dataset (2 years) | √ | | √ | √ | √ | √ (T) | √ (CS) | N/A |
| Corter, 2007 [60] Corter, 2008 [62] Corter, 2009 [61] Corter, 2012 [63] | **Toronto First Duty** (low) | Small and variably defined | Secondary datasets (4 years) | √ | | √ | | √ | √ (G) | √ (CS) | N/A |
| NNI Research Team, 2007 [47] | **Neighbourhood Nurseries Initiative** (medium) | N/A | Study designed and collected data, and secondary datasets (4 years) | √ | | | | √ | √ (T+G) | X (CS) | N/A |
| McKeown, 2014 [64] | **NEYAI** (low) | Small and variably defined | Study designed and collected data (2.5 years) | √ | | √ | √ | √ | X | N/A | N/A |
| Browning, 2010 [55] Compass Evaluation & Research, 2015 [56] | **First Steps** (medium) | Large and uniformly defined | Study designed and collected data, along with secondary datasets (6 years) | √ | √ | √ | √ | √ | X | N/A | N/A |
| Mackenzie, 2004 [66] Shute, 2005 [65] | **Starting Well** (low) | Small and uniformly defined | Secondary datasets (2 years) | √ | √ | √ | √ | √ | X | N/A | N/A |
| **Studies with no comparison group** | | | | | | | | | | | |

(*Continued*)

**Table 5.** (Continued)

| First author(s), year | Initiative (quality rating) | Size of local delivery areas | Sample data for impact evaluation (study length) | Mechanisms by which child and family outcomes will be achieved | | | | | Evidence of positive impact on *pregnancy/birth/child* outcomes at population level | | |
|---|---|---|---|---|---|---|---|---|---|---|---|
| | | | | Enhance, intensify, collocate or redesign universal services | Address unmet service gaps | Joined up working / collaboration | Community involvement | Local discretion /variation | Measured at population level? | Evidence of impact at population level? | Evidence of sustained impact? |
| | | | | | | | | | General (G) or Targeted (T) sample | Cohort (CO) or Cross-sectional (CS) | |
| Hickey, 2018 [40] | **ABC Programme** (low) | Small and variably defined | Study designed and collected data (3 years) | √ | | √ | √ | √ | X | N/A | N/A |

intervention sample (participants who were known to have used the intervention) there was positive impact on work status and childcare uptake. Their examination of reach found that the initiative only reached 10% of the eligible population.

There is no clear relationship between the size of the local delivery area and initiative effectiveness, with initiatives implementing 'local' solutions at a large (e.g., county) and small (e.g., school neighbourhood) sized area demonstrating impact. Nor is there a clear relationship between the mechanisms by which the intervention was theorised to improve outcomes and effectiveness, although the inclusion of universal services (maternal and child health services, childcare, pre-school) in the service model of initiatives appeared to be mostly beneficial in demonstrating positive results.

## Discussion

In this review, we examined the evidence for the effectiveness of public policy driven place-based initiatives for children, while also examining the study designs and methods used to evaluate the initiatives, and the context in which the initiatives were implemented and evaluated. The initiatives identified were diverse in their service delivery, evaluation designs and the range and number of outcomes assessed. Most were of medium-quality for evaluating place-based initiatives. Key findings and recommendations for policy makers and evaluators are discussed below.

While RCTs are considered the gold standard for assessing the effectiveness of single, well-defined interventions, such approaches are less appropriate for large complex public health interventions [73]. In assessing the study designs and methods employed (Aim 1), we found the vast majority of initiatives reviewed here employed quasi-experimental designs, with considerable variability in the sampling methods. As place-based initiatives aim to impact on whole-of-community outcomes, impact studies should use community-level samples, not samples of those who receive specific services ('intervention samples'). General population samples may be appropriate for initiatives that are more prescriptive with a common set of outcomes to be achieved by all local areas. For initiatives with a high degree of local flexibility, using a 'targeted' population sample is more appropriate, whereby an outcome of interest is assessed

only within the communities where that outcome was explicitly prioritised and targeted (as used by Best Start [52–54] and Georgia Family Connection [59]). In practice this means designing rigorous data collection systems that enable the 'filtering' of outcome measure evaluation to include only those local areas that targeted that outcome measure specifically.

An intervention sample design that only includes those who have been exposed to specific services or programs is a weak study design for the evaluation of place-based initiatives and should not be used. Place-based initiatives are intended to improve whole communities and all people living in them (the 'neighbourhood effect' or community-level change), not just those receiving some form of the intervention. Initiatives that measure at a sample-level only are more likely to have positively skewed results and should be regarded with caution.

While some place-based initiatives have study-designed long-term impact studies, these are difficult to sustain due to cost, participant attrition, and the difficulty maintaining the integrity of suitable comparison areas [44, 74]. Many of the studies examined here assessed long-term outcomes by analysing routinely collected datasets. However, this approach has the disadvantage of outcome measures being selected from what is available rather than what is ideal [74], and may result in a misestimation of effectiveness. A longitudinal impact evaluation with multiple follow-up points is the optimal method for measuring the effectiveness of place-based initiatives. Routinely collected datasets and mechanisms for linkage are becoming increasingly available through governments in Australia and elsewhere. These provide the most promising way forward for future study designs. Time trend studies can also provide critical evidence of the long-term impact of place-based initiatives and their use should be explored further. A recent time trend study of the long term impact of the UK Labour government's 1997–2010 strategy to reduce geographic health inequalities (that included Sure Start) found the strategy substantially reduced inequalities, compared with trends before and after the strategy [11]. The authors noted that previous studies evaluating components of the strategy had found weak evidence of impact.

Our review found many elements of process evaluation were not examined, reflecting inherent difficulties in trying to assess service offerings that may vary considerably at the local level. Wilks and colleagues similarly found that many of the elements common to place-based initiatives were not evaluated [6]. Nevertheless, a clear process evaluation framework, linked to an initiative's theory of change, should be conceived and executed to determine whether initiatives are implemented as intended, as this has important implications for their effectiveness [75]. Local evaluations are one part of the solution [13, 17], but require expert guidance and support [16]. Dedicated and sufficient funding should be allocated to local evaluation to ensure service providers can source such support and build local capacity. Local evaluation findings need to be consolidated at the whole-of-initiative level, and while this is challenging, others have provided recommendations for streamlining this process [13, 17]. These 'local lessons' are too important to lose.

It was notable in our review that for most initiatives, the commissioning and design of an evaluation occurred after implementation had commenced. O'Dwyer and colleagues [23] made a similar finding. This can significantly restrict the methods able to be employed, limiting the value of evaluation [75]. Of particular concern, pre-intervention baseline data were not available for many of the initiatives assessed here. Evaluation frameworks should be designed at the same time as the design of the initiative and in place prior to the commencement of implementation. This is an important recommendation for those commissioning place-based initiatives.

Place-based initiatives need sufficient lead time to develop and implement interventions in each community before whole-of-initiative effects can expect to be observed. Place-based interventions require service providers at a local level to scale up and implement new

programs and services to make use of the funding available to them. This can take considerable time, particularly in regional and remote areas where infrastructure is spare, where recruitment of suitably qualified personnel takes time, and where new partnerships need to be established and embedded. Yet governments want to see quick results, and investment beyond a few years is uncommon. Rae [76] suggests that these types of policy approaches should be considered a 25 year investment. Additionally, some benefits for disadvantaged children do not become apparent until they have reached adulthood [77–79]. The systematic review of place-based initiatives to reduce health inequalities conducted by O'Dwyer and colleagues [23] found four of 24 initiatives reviewed were evaluated three years after implementation. The present review differs in that multiple evaluations of the same initiative were combined and we examined the final time participants were followed up, yet we found a similar lack of long-term evaluations. Evaluating for impact should be planned but not commence until at least three years after an initiative has been established and is fully operational.

Our second aim was to examine the context in which the initiatives were implemented and evaluated. We looked for social, political and economic factors affecting the delivery and evaluation of initiatives. With the exception of time-limited demonstration projects, many initiatives were subject to changes in funding, scope or design of the initiative and/or evaluation. In some cases the evaluators of these initiatives theorised how changes might impact outcomes, while in others they were largely silent. Context is an active rather than a passive construct, which "...interacts, influences, modifies and facilitates or constrains..." interventions [80, section-17-1-2-1], and the contextual changes we observed are almost inevitable with long-term public policy initiatives. Thus contingency planning is required from the outset, along with a rigorous assessment of their impact on implementation and outcomes. Frameworks that take into account context in implementation of complex interventions can help [81].

Our third aim was to evaluate the effectiveness of place-based initiatives in improving outcomes for children. While all assessed initiatives were able to demonstrate at least one positive benefit, the initiatives used a broad range of measures and at several time points did not demonstrate widespread sustained positive benefits [39, 45]. This is consistent with the findings of other reviews of place-based initiatives [20, 21, 23]. Possible explanations have been discussed above but are summarised again here: poor study design (in terms of sampling, measurement selection and timing); the selection of different target outcomes at a local area level diluting the capacity to detect whole-of-initiative level change; initiatives not implemented as intended; and the influence of changing contextual factors over time. All of these were present in the initiatives reviewed here. The heterogeneity of the initiatives' design, objectives, theories of change, size of delivery area, service model, implementation and outcomes made it difficult to draw conclusions about what aspects contributed to positive benefits where they were demonstrated. Lack of attention to 'place' in some initiatives may have also impacted their effectiveness and was noted in the consolidated local evaluation reports examined in this review. Understanding and evaluating the local variability in intervention areas, and how services and the community interact with each other and with neighbouring services is a consideration that requires further exploration [6, 23].

This review identified a broad range of child outcomes measured across the 12 initiatives, reflecting the varying initiative objectives, settings and data available for measurement at the time they were established and evaluated. Given this heterogeneity, we recommend all child-focused place-based initiatives use a core set of indicators such as those established by the United Nations Sustainable Development Goals. There are now 35 agreed outcome indicators directly related to the health and wellbeing of children, in areas such as poverty, health and wellbeing, and education, many covering early childhood development [82]. Incorporating at least some of these child outcome domains would help to achieve consistency in measurement and allow comparison and synthesis of child outcome data across studies.

### Limitations and directions for future research

This review was subject to some limitations. We excluded philanthropic and community-led initiatives, reflecting the priorities of the research team and also the pragmatic challenges associated with systematically identifying literature relevant to these initiatives which are often dispersed across multiple reports in the grey literature. As the search was on English language papers only, there may be European initiatives that were excluded. There are numerous protocols and process evaluation studies of place-based initiatives, and some impact studies, including several in Europe which did not meet the criteria for inclusion [83–85]. The heterogeneity of the studies included meant it was not possible to conduct a statistical meta-analysis of outcome data and there was insufficient commonality for us to meaningfully summarise sub-group analyses.

Limited research has been conducted into the impact of scope or design changes. For example, three initiatives included in this review introduced a requirement to use evidence-based programs. This was hypothesised as positive and beneficial for children and families, however others have suggested that the mandated use of evidence-based programs does not always have the intended effect and has unintended consequences at a local level [86, 87]. Little is known about the knowledge and experiences of personnel implementing mandated evidence-based programs in place-based initiatives. The influence of top-down changes such as these is an area of research requiring further study.

## Conclusion

Despite the growth of place-based initiatives to improve outcomes for children residing in disadvantaged areas, the evidence for the effectiveness of such initiatives remains unconvincing, which may reflect a failure of the evaluation designs or a failure of the initiatives themselves. Power and colleagues [20] have suggested that the blindness of governments to the underlying structural inequalities in our societies means that place-based initiatives will do little more than nudge at the margins of change. Similarly, Bambra and colleagues [88] suggest that macro political and economic structures have a far greater influence on geographical inequalities than local environments. Others have suggested that while the theory underpinning place-based approaches is sound, issues such as poor problem conceptualisation, lack of understanding of the spatial scale of problems, and initiatives overreaching relative to their funding and timeframes means successful initiatives are rare [21, 76]. The authors of the present review fall into the latter camp. We remain optimistic on the basis that some positive effects have been found despite the many evaluation design limitations. We are disappointed however, that the lessons learned in earlier evaluations and literature reviews have not been acted on, and the same mistakes are being made time and time again. What is critical going forward, is greater investment and planning in evaluation to avoid the absence of quality effectiveness data from being interpreted as an absence of effectiveness, and being used to justify the defunding of place-based initiatives.

## Supporting information

**S1 Appendix. Example search strategy–EMBASE.**
(DOCX)

**S2 Appendix. Quality of impact study based on fit-for-purpose.**
(DOCX)

**S3 Appendix. Tables of study reported outcomes by categories and domains.**
(DOCX)

**S1 Checklist. Preferred reporting items for systematic reviews and meta-analyses extension for scoping reviews (PRISMA-ScR) checklist.**
(DOCX)

## Acknowledgments

Our appreciation and thanks go to Professor Donna Berthelsen (School of Early Childhood and Inclusive Education, Queensland University of Technology), for her wisdom and advice.

## Author Contributions

**Conceptualization:** Fiona C. Burgemeister, Sharinne B. Crawford, Naomi J. Hackworth, Jan M. Nicholson.

**Data curation:** Fiona C. Burgemeister.

**Formal analysis:** Fiona C. Burgemeister, Jan M. Nicholson.

**Methodology:** Fiona C. Burgemeister.

**Supervision:** Sharinne B. Crawford, Naomi J. Hackworth, Stacey Hokke, Jan M. Nicholson.

**Validation:** Jan M. Nicholson.

**Visualization:** Fiona C. Burgemeister.

**Writing – original draft:** Fiona C. Burgemeister.

**Writing – review & editing:** Fiona C. Burgemeister, Sharinne B. Crawford, Naomi J. Hackworth, Stacey Hokke, Jan M. Nicholson.

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
