## [Decision Letter · Decision Letter 0]

8 Jun 2021

PONE-D-20-30217

Place-based approaches to improve health and development outcomes in young children: a systematic review

PLOS ONE

Dear Dr. Burgemeister,

Thank you for submitting your manuscript to PLOS ONE. After careful consideration, we feel that it has merit but does not fully meet PLOS ONE’s publication criteria as it currently stands. Therefore, we invite you to submit a revised version of the manuscript that addresses the points raised during the review process.

Please note that your manuscript was reviewed by 2 experts in the field . There is consensus agreement that the idea of the article is interesting but also consensus agreement that the article required additional work. The reviewers identified important problems and provided copious comments (enclosed).

The manuscript could be greatly strengthened by considering editing according to the specific Reviewers’ comments.

Please submit your revised manuscript by July 21 2021 11:59PM. If you will need more time than this to complete your revisions, please reply to this message or contact the journal office at plosone@plos.org. Please include the following items when submitting your revised manuscript:

We look forward to receiving your revised manuscript.

Kind regards,

Ammal Mokhtar Metwally, Ph.D (MD)

Academic Editor

PLOS ONE

Journal Requirements:

2) Please include captions for your Supporting Information files at the end of your manuscript, and update any in-text citations to match accordingly. Please see our Supporting Information guidelines for more information: http://journals.plos.org/plosone/s/supporting-information.

Reviewers' comments:

Reviewer's Responses to Questions

**Comments to the Author**

1. Is the manuscript technically sound, and do the data support the conclusions?

Reviewer #1: Yes

Reviewer #2: Partly

2. Has the statistical analysis been performed appropriately and rigorously? 

Reviewer #1: Yes

Reviewer #2: N/A

3. Have the authors made all data underlying the findings in their manuscript fully available?

Reviewer #1: Yes

Reviewer #2: Yes

4. Is the manuscript presented in an intelligible fashion and written in standard English?

Reviewer #1: Yes

Reviewer #2: Yes

5. Review Comments to the Author

Reviewer #1: This Systematic Review has been carefully conducted with strict adherence to the PRISMA principles. The conclusions reached are plausible. It also clearly brings out the varied problems with studies evaluating the place-based approaches to improve health and development outcomes in developed countries

Reviewer #2: PONE-D-20-30217

Stated aims of this manuscript:

This review aims to examine the strength of the evidence for the effectiveness of government-led place-based interventions to improve outcomes for disadvantaged children, families and communities [per abstract].

"this study focuses on early childhood initiatives that target..pregnancy to 4 years." Lines 101-103:

Comments:

An interesting investigation based on much time and effort. Please excuse any repetition below.

My overall assessment is that this is a scoping vs systematic review; that the inclusion criteria--including population and outcomes, need to be more fully and clearly stated such that the results reported are in better alignment. Tables and the text should be better referenced, e.g. Tables do not have references. Given the (very good and much appreciated) approach of collecting multiple reports of single studies/initiatives, a table showing references grouped by initiative would be useful. Take a look at an included studies table in a Cochrane Systematic review to see an example if my suggestion is not clear. Based on above excerpts, one from the abstract, one from the text, I am not sure the aim of the review is stated consistently.

RE type of review: I can see that a number of standard systematic review processes were followed, e.g. documented literature search; quality assessment, but these procedures are also part of scoping review methodology. The inclusion of any study design supports, I think, the idea that this is a scoping review, as does the use of single (unvalidated?) quality assessment checklist across multiple designs (line 192). There are, as you know, standard and validated Risk of Bias and quality assessment tools for a variety of study designs and use of such tools is part of systematic review methodology. I notice the authors reference "A schema for evaluating evidence of public health interventions' and this source provides guidance on assessing studies based on design. In this review, studies with similar designs do not appear to be grouped together and analysed--which may have been possible since I am uncertain as to whether heterogeneity referred to all studies, regardless of design or whether it was found within studies using the same design. Further, in systematic reviews, quality is less the issue than risk of bias inherent in various study designs. Issues of quality are typically dealt with by using GRADE in conjunction with RoB in systematic reviews in order to provide an estimation of the strength of evidence by outcome and across studies.

This article about scoping reviews references a number of the earlier/foundational papers on the method: https://www.journalofhospitalmedicine.com/jhospmed/article/202729/hospital-medicine/methods-research-evidence-synthesis-scoping-review and may provide an indication of my opinion regarding my assessment that this is a scoping vs systematic review.

I believe it would be useful to better delineate inclusion criteria following, roughly PICOS-ish categories; a clear delineation of outcomes or outcome data of interest would, for instance be quite useful and would better prepare the reader for the results sections.

I believe some of the information that seems either under-explained or missing from the text of the review is found in Tables, but I would like the Tables to support the text vs having to scrutinize the tables to better understand the narrative description of findings.

A number of detailed remarks below.

Table 2: I'm not sure i understand the numbers reported for Peer reviewed only; Grey only; and Mixed.

LIne 193-194: I'd be interested to see references here to support the statement re "standard evaluation methods..."

Line 380; I believe references should be provided here so we know which 12 studies are being discussed; same comment for lines 385-386, 402. etc.

Table 4: I believe there should be a column for design vs relying on Comparison group and no comparison group sub-sections. I'd also like to see references in this and other tables.

PRISMA is referred to as the guide for the review, but PRISMA tells authors what to report, not how to "do" or conduct a systematic review; referencing PRISMA in this way is common across the literature, but it is incorrect. The Cochrane Handbook is an example of a source for methodological guidance and I believe it is cited in the reference list of this review; if it was the guide of the authors' methods, I'd suggest stating this in the text. Having said this, referencing methodological guidance on scoping reviews may be more appropriate.

Search and Search Methods: Quite well done; especially in the choice of multiple databases. However, I believe it would have been worthwhile to explore controlled vocabulary in at least Embase and Medline/Pubmed where there are thesauri. Regarding databases: ProQuest and Informit are platforms and/or database providers. Both ProQuest and Informit provide access to many databases. I notice that Informit, for instance, offers the Australian Education Index and Australian Public Affairs Information Service, to name two. Were searches in ProQuest and Informit simply done across multiple databases? If so, make this clear, and perhaps name a few of the most relevant (to this review) databases from each database provider; if, on the other hand, only specific ProQuest and Informati databases were searched, these should be named.

Terminology:

I'd suggest referring to the current work as a "review" vs study; use the word study to describe the included studies. For reference, there is a section, pg 4, called "The Current Study."

Peer reviewed. I am not sure this phrase or criteria should be used--especially since grey literature reports--which are often not peer reviewed--were sought during the search.

In Eligibility Criteria, one subheading title is 'Initiative selection". It might be a bit clearer to either omit this subtitle, or rename it inclusion criteria. It might also be useful to use bullets or brief, descriptive phrases prior to the description of each criteria, e.g.

Item 1: Population: Socially disadvantaged children (infancy to 4 years) and pregnant women.

Item 2: Location: high-income countries

Sponsoring organization (or something like that): governmentally administered program. The question of the level of government seems more an outcome.

Number of sites/extent of program: more than one site

Item 4 seems to repeat information in item 3 where level of government is mentioned.

Item 5: Reported program evaluation examining more than one child-focussed outcome.

Lines 153 and 154 are a bit unclear; see above suggestion to clarify inclusion criteria, eg "more than one...outcome"

I think a subheading for this paragraph, Exclusion Criteria, would be useful.

Line 159: Article Selection

Here I would describe, using standard systematic review language, the Screening Process. E.g presumably title/abstracts were screened first with eligible or potentially eligible studies moved forward for full-text review. Was screening undertaken independently by two authors? I suggest clearly stating the process.

Lines 160 Re. the distinction between Primary and Secondary "articles" suggests you have taken a "study based" approach, similar to that used by Cochrane reviews where multiple references for a given study are collated and examined for data. This is a very good idea and demonstrates your thoroughness.

Line 163 refers to "additional articles"; use the word "study" or "study reports"; I'd also suggest sticking with the primary/secondary terminology.

I would like to see a section title Outcomes where the outcomes are detailed. On Page 18, summaries of the Outcomes or Data of interest begin, e.g. Size and selection of delivery areas; theories of change... but I'd like to see these delineated earlier. Table 1 mentions outcome domains, but these do not seem to correspond to the narrative summary beginning on pg 18.

LIne 171: "articles were excluded if..." This is an eligibility criteria; I suggest adding this information to the Eligibility Section by creating a sub-heading(s) for "publication type" and "study design" where you would state, positively, the designs you considered eligible and ineligible; it would be worthwhile to describe clearly acceptable designs, e.g. pilot studies; pre-post, cohort, controlled, etc. Line 171 also suggests you included "reviews"--I would like to see this defined because a "review" does not sound like a "study." Be specific in the Inclusion/Eligibility section vs later stating what you excluded.

Line 172: here you are restating what should be clear from the Eligibility section, eg. of course you are excluding programs that are not place-based initiatives.

Line 173: what is a "local evaluation"?

Line 173-174: duplication of information in a higher quality source". I do not understand this and, systematic reviews do not exclude studies based on "quality" or high risk of bias; instead, these weaknesses are taken into account in statistical or narrative synthesis.

Line 174-175: All publications...this is a clear statement, but should be positioned earlier in the paragraph when describing how you grouped/collated multiple reports of the same study.

Table 1, Line 190: I'm not sure what the "ratings" in the title of this table refer to.

line 310: Evaluation Designs. I believe this section tries to break out and describe different approaches to evaluation methods or models used in the included studies, have I got that right? If so, then I would say the language and headings in this section need reworking. Based on line 317 it sounds like program evaluations may be multi-faceted and include various approaches or elements such as impact evaluation, process evaluation, local evaluations, and cost analyses. Earlier in the paper I do not see the phrases referring to these elements. I also think these elements should be briefly defined, especially if subsequent headings refer to them. I notice in a note to Table 3 the terms impact/process etc are mentioned, but explication in the text of the review would be helpful.

Line 379 "impact (or outcome) evaluation studies varied considerably in design". Does this mean that "impact studies" are a type of study, or is "impact evaluation" a characteristic of program evaluation?

6. PLOS authors have the option to publish the peer review history of their article (what does this mean?). If published, this will include your full peer review and any attached files.

Reviewer #1: No

Reviewer #2: **Yes: **Michelle Fiander

---

## [Author Response · Author response to Decision Letter 0]

19 Aug 2021

Responses to specific reviewer and editor comments have been provided in a document attached to this submission. We would like to thank both reviewers for their time and feedback on this paper. We feel the thorough review of this manuscript and feedback we received have significantly strengthened it.

---

## [Decision Letter · Decision Letter 1]

15 Oct 2021

PONE-D-20-30217R1Place-based approaches to improve health and development outcomes in young children: a scoping reviewPLOS ONE

Dear Dr. Burgemeister,

Thank you for submitting your manuscript to PLOS ONE. After careful consideration, we feel that it has merit but does not fully meet PLOS ONE’s publication criteria as it currently stands. Therefore, we invite you to submit a revised version of the manuscript that addresses the points raised during the review process.

We look forward to receiving your revised manuscript.

Kind regards,

Ammal Mokhtar Metwally, Ph.D (MD)

Academic Editor

PLOS ONE

Journal Requirements:

Additional Editor Comments (if provided):

Great effort has been made by the authors to utilize the feedback that was provided for them to correct. I find it interesting and improved with respect to the original submission. I really encourage the authors to conduct a similar study targeting low- and middle-income countries highlighting the best practices according to their sociodemographic context. However, there are few minor things to adjust related to the introduction section which is lengthy. The authors should consider revising it without putting subheading.

Reviewers' comments:

Reviewer's Responses to Questions

**Comments to the Author**

1. If the authors have adequately addressed your comments raised in a previous round of review and you feel that this manuscript is now acceptable for publication, you may indicate that here to bypass the “Comments to the Author” section, enter your conflict of interest statement in the “Confidential to Editor” section, and submit your "Accept" recommendation.

Reviewer #1: All comments have been addressed

Reviewer #2: All comments have been addressed

2. Is the manuscript technically sound, and do the data support the conclusions?

Reviewer #1: Yes

Reviewer #2: Yes

3. Has the statistical analysis been performed appropriately and rigorously? 

Reviewer #1: Yes

Reviewer #2: N/A

4. Have the authors made all data underlying the findings in their manuscript fully available?

Reviewer #1: Yes

Reviewer #2: Yes

5. Is the manuscript presented in an intelligible fashion and written in standard English?

Reviewer #1: Yes

Reviewer #2: Yes

6. Review Comments to the Author

Reviewer #1: The manuscript, though long with rather large tables, reasonably addresses the questions set in the objectives.

Reviewer #2: Thank you for considering my suggestions. I think the additional headings to introduce the various aspects of your conceptual model and findings is helpful to readers.

7. PLOS authors have the option to publish the peer review history of their article (what does this mean?). If published, this will include your full peer review and any attached files.

Reviewer #1: No

Reviewer #2: **Yes: **Michelle Fiander

---

## [Author Response · Author response to Decision Letter 1]

28 Oct 2021

Responses to reviewer and editor comments are contained in the Word document attached.

---

## [Decision Letter · Decision Letter 2]

11 Nov 2021

PONE-D-20-30217R2Place-based approaches to improve health and development outcomes in young children: a scoping reviewPLOS ONE

Dear Dr. Burgemeister,

Thank you for submitting your manuscript to PLOS ONE. After careful consideration, we feel that it has merit but does not fully meet PLOS ONE’s publication criteria as it currently stands. Therefore, we invite you to submit a revised version of the manuscript that addresses the points raised during the review process. Please consider including your revision within your response to reviewer comments inside the document of "response to reviewers' comments" to facilitate the understanding of the changes that have been made by the reviewers. In other words, any revision or added references should be included in the document "response to reviewers' comments" and not only as track changes

We look forward to receiving your revised manuscript.

Kind regards,

Ammal Mokhtar Metwally, Ph.D (MD)

Academic Editor

PLOS ONE

Journal Requirements:

Reviewers' comments:

Reviewer's Responses to Questions

**Comments to the Author**

1. If the authors have adequately addressed your comments raised in a previous round of review and you feel that this manuscript is now acceptable for publication, you may indicate that here to bypass the “Comments to the Author” section, enter your conflict of interest statement in the “Confidential to Editor” section, and submit your "Accept" recommendation.

Reviewer #1: All comments have been addressed

Reviewer #2: All comments have been addressed

2. Is the manuscript technically sound, and do the data support the conclusions?

Reviewer #1: Yes

Reviewer #2: Yes

3. Has the statistical analysis been performed appropriately and rigorously? 

Reviewer #1: Yes

Reviewer #2: I Don't Know

4. Have the authors made all data underlying the findings in their manuscript fully available?

Reviewer #1: Yes

Reviewer #2: Yes

5. Is the manuscript presented in an intelligible fashion and written in standard English?

Reviewer #1: Yes

Reviewer #2: Yes

6. Review Comments to the Author

Reviewer #1: The authors have addressed the issues raised in earlier reviews satisfactorily. The data is clearly presented and description of the tables is easy to follow

Reviewer #2: I do not need to see or comment on this review again, but have a final few comments--the authors can take or leave. Congratulations on completing this!

Best wishes,

Michelle Fiander

Abstract

Changes suggested here, if accepted, should be made throughout the manuscript.

" Methodologies and methods for evaluating such place-based initiatives were assess" --I am not sure how methodologies and methods are being distinguished here. Suggest clarifying here and anywhere else these two terms are used. BY methodology do you mean study design?

You may want to change the word "records" to "reports" ; e.g. 32 reports relating to 12 initiatives.

"Many initiatives were affected by external factors such as policy and funding changes, with unknown impact" This is a finding and if the statement stays in the abstract, it should be repositioned--making it the second last sentence seems the right place. Having said this, I'd say the statement requires clarification. I'm left asking: did these factors mean outcomes couldn't be measured in the studies? Even when I reword the sentence: "External factors such as policy and funding changes had an unknown impact on...." I wonder what I'm meant to understand.

Use of the word "significant". I'd be inclined to remove this adjective. You provide the stats so let them speak for themselves--especially when there is no a priori standard of what is or is not significant change.

Line 33: I would state the number of studies with comparison groups; you mention 6 studies reported more than 1 post baseline measure; how many studies reported only 1 post baseline measure? did you only include data when there was 1+ post measure?

Line 101: remove conceived; I think all you can say is that some studies describe the intervention as place-based, while others discuss "places" in terms of neighbourhoods, geography or location.

7. PLOS authors have the option to publish the peer review history of their article (what does this mean?). If published, this will include your full peer review and any attached files.

Reviewer #1: No

Reviewer #2: **Yes: **Michelle Fiander

---

## [Author Response · Author response to Decision Letter 2]

12 Nov 2021

Responses to reviewer comments are provided in the document attached.

---

## [Editor Report · Decision Letter 3]

15 Nov 2021

PONE-D-20-30217R3Place-based approaches to improve health and development outcomes in young children: a scoping reviewPLOS ONE

Dear Dr. Burgemeister,

Thank you for submitting your manuscript to PLOS ONE. After careful consideration, we feel that it has merit but does not fully meet PLOS ONE’s publication criteria as it currently stands. Therefore, we invite you to submit a revised version of the manuscript that addresses the points raised during the review process.

We look forward to receiving your revised manuscript.

Kind regards,

Ammal Mokhtar Metwally, Ph.D (MD)

Academic Editor

PLOS ONE

Journal Requirements:

Additional Editor Comments (if provided):

This study highlights on the “Place-based approaches to improve health and development outcomes in young children: a scoping review”

Unfortunately, it seems that there is a problem all through the manuscript concerning the focus of the study in term of the measured outcome of health and development of the children. This should be clear in the abstract and introduction not only in the methodology section

As you know health and development outcome is a very broad aspect and it was expected from the authors to clearly indicate in their abstract what aspects of health and development outcome the study focused. Please consider to be focused and concise in relation to the following: Aim (This scoping review examines the strength of evidence for the effectiveness of public policy-led place-based initiatives designed to improve outcomes for disadvantaged children, their families and the communities in which they live (be specific: which outcome?)

Methodology: The authors mentioned”. Eleven initiatives used a quasi-experimental evaluation to assess impact, although there were considerable design variations within this…..” (impact on what….). The authors have to mention in their design the aspects of the impact they have targeted in their study.

The results: The authors mentioned” Across the 83 outcome domains reported in the 11 studies with a comparison group, 30 (36.4%) demonstrated a positive outcome, and all but one initiative demonstrated a positive outcome in at least one outcome measure. Of the six studies that examined outcomes more than once post baseline, 10 from 38 outcomes (26.3%) demonstrated positive sustained results. (Again, what were theses outcomes?)

Introduction: line 49, 51,52 again, please specify which outcome (for the health aspects as well as the developmental domains) What were the study looking for are not clear

Line 61 what do you mean by (7, p21)! You should not mention the page number.

Again line 64: which outcome (s) the study addressed?

Line 91, 92: The authors mentioned “There is no contemporary literature review that examines evidence of the effectiveness of place-based initiatives for children in their early years” be specific again …. Initiatives for which outcome,

Line 114 to 115 : The authors mentioned “This review focuses on early childhood initiatives that target (but are not necessarily limited to) pregnancy to four years”. Again be specific.

Methodology: There is a lot of recent interventions studies in middle east region targeted child outcome in general and development the majority of which were community based or facility based and in rural communities,…… that fit and the authors did not mention any. Please consider adding these studies to enrich your methodology and your review.

The authors considered exactly the exclusion criteria but did not mention in the inclusion criteria what are the combined outcomes they focused on (please consider this)
---

## [Author Response · Author response to Decision Letter 3]

21 Nov 2021

The response to editor and reviewer comments is provided in the submitted document attached.

---

## [Editor Report · Decision Letter 4]

9 Dec 2021

Place-based approaches to improve health and development outcomes in young children: a scoping review

PONE-D-20-30217R4

Dear Dr. Burgemeister,

We’re pleased to inform you that your manuscript has been judged scientifically suitable for publication and will be formally accepted for publication once it meets all outstanding technical requirements.

Kind regards,

Ammal Mokhtar Metwally, Ph.D (MD)

Academic Editor

PLOS ONE
---

## [Editor Report · Acceptance letter]

14 Dec 2021

PONE-D-20-30217R4 

Place-based approaches to improve health and development outcomes in young children: a scoping review 

Dear Dr. Burgemeister:

I'm pleased to inform you that your manuscript has been deemed suitable for publication in PLOS ONE. Congratulations! Your manuscript is now with our production department. 

Kind regards, 

on behalf of

Professor Ammal Mokhtar Metwally 

Academic Editor

PLOS ONE